# H7N9 avian influenza virus infection in men is associated with testosterone depletion

Tian Bai [1,2,10], Yongkun Chen [3,4,10], Sebastian Beck[1,10],
Stephanie Stanelle-Bertram[1], Nancy Kouassi Mounogou[1], Tao Chen[2], Jie Dong[2],
Bettina Schneider[5], Tingting Jia[3,4], Jing Yang[2], Lijie Wang[2], Andreas Meinhardt[6],
Antonia Zapf[7], Lothar Kreienbrock[5], Dayan Wang[2], Yuelong Shu [2,3,4,8,11] ✉ &
Gülsah Gabriel [1,9,11] ✉

Human infections with H7N9 avian influenza A virus that emerged in East China in 2013 and caused high morbidity rates were more frequently detected in men than in women over the last five epidemic waves. However, molecular markers associated with poor disease outcomes in men are still unknown. In this study, we systematically analysed sex hormone and cytokine levels in males and females with laboratory-confirmed H7N9 influenza in comparison to H7N9-negative control groups as well as laboratory-confirmed seasonal H1N1/H3N2 influenza cases (*n* = 369). Multivariable analyses reveal that H7N9-infected men present with considerably reduced testosterone levels associated with a poor outcome compared to non-infected controls. Regression analyses reveal that testosterone levels in H7N9-infected men are negatively associated with the levels of several pro-inflammatory cytokines, such as IL-6 and IL-15. To assess whether there is a causal relationship between low testosterone levels and avian H7N9 influenza infection, we used a mouse model. In male mice, we show that respiratory H7N9 infection leads to a high viral load and inflammatory cytokine response in the testes as well as a reduction in pre-infection plasma testosterone levels. Collectively, these findings suggest that monitoring sex hormone levels may support individualized management for patients with avian influenza infections.

In early 2013, human infections with avian influenza A (H7N9) virus were first reported in China[1]. Since then, H7N9 caused five epidemic waves from 2013 to 2017, making it one of the major zoonoses in the current decade. Exposure to live poultry markets is considered a major risk factor for human infections with the H7N9 virus[2]. An initial

epidemiological study revealed that 71% of all H7N9 cases were males in the first wave[3]. A high degree of infection in men was consistently observed, with similar proportions (68–71%) throughout the entire five H7N9 epidemic waves[4]. Currently, very little is known regarding the complex interplay of biological sex and gender-specific behaviour as

[1]Viral Zoonoses-One Health, Leibniz Institute for Virology (LIV), Hamburg, Germany. [2]Chinese National Influenza Center, National Institute for Viral Disease Control and Prevention, Chinese Center for Disease Control and Prevention, Beijing 102206, P.R. China. [3]School of Public Health (Shenzhen), Sun Yat-sen University, Guangdong 510275, P.R. China. [4]School of Public Health (Shenzhen), Shenzhen Campus of Sun Yat-sen University, Shenzhen 518107, P.R. China. [5]Department of Biometry, Epidemiology and Information Processing, University of Veterinary Medicine Hannover, Hannover, Germany. [6]Institute for Anatomy and Cell Biology, Justus-Liebig University of Gießen, Gießen, Germany. [7]Institute for Medical Biometry and Epidemiology, University Medical Center Hamburg-Eppendorf, Hamburg, Germany. [8]Institute of Pathogen Biology, Chinese Academy of Medical Sciences & Peking Union Medical College, Beijing, China. [9]Institute of Virology, University of Veterinary Medicine, Hannover, Germany. [10]These authors contributed equally: Tian Bai, Yongkun Chen, Sebastian Beck. [11]These authors jointly supervised this work: Yuelong Shu, Gülsah Gabriel. ✉e-mail: shuylong@mail.sysu.edu.cn; guelsah.gabriel@leibniz-liv.de

contributors to influenza disease outcomes. The World Health Organization (WHO) first proposed that gender-associated practises and norms, such as more frequent exposure of men to birds, may pose one of the reasons for sex-specific H7N9 incidence[5]. However, a follow-up study suggested that an increased risk in older men is not due to higher exposure time in the live poultry market[6]. Thus, the impact of biological sex on H7N9 influenza outcome remains unknown.

Sex hormones, particularly testosterone and estradiol, are important biological factors that affect sex differences in immune responses[7–10]. We have previously shown that high-dose H1N1 infection reduces testosterone levels in male (but not female) mice[11]. Conversely, treatment of female mice with testosterone protects the majority of H1N1-infected females from lethal outcomes[11]. Similarly, aged male mice with low testosterone levels were reported to undergo elevated pulmonary inflammation and severe disease upon H1N1 influenza virus infection compared to young male mice[12]. Conversely, other murine models showed that low concentrations of estradiol in female mice were associated with enhanced pro-inflammatory responses and H1N1 influenza pathogenesis[13]. Thus, there is increasing evidence from small animal models that sex hormones play an important role in influenza virus pathogenesis. However, the role of sex hormones in infectious diseases in humans remains to be elucidated.

In this study, we retrospectively analyse the association of sex hormones with H7N9 influenza disease outcome in males and females and show that low testosterone levels are linked to the development of severe or even fatal diseases. Moreover, in a mouse model of H7N9 infection, we demonstrate a causal link between viral infection and testosterone depletion. In summary, our data suggest that reduced testosterone levels upon H7N9 infection represent a poor prognostic marker in critically ill patients. Monitoring sex hormone levels throughout the course of infection might therefore be essential for risk assessment and individualized patient therapy.

## Results

### Participant demographics

In total, we enrolled $n = 369$ study participants, which were subdivided into those 18–49 years old (reproductive age) and those 50 years of age and older (post-reproductive age) to adequately assess sex hormone levels with respective control groups (Table 1). In both age groups, patients were enrolled with PCR-confirmed H7N9 influenza A virus (IAV) infection (within acute phases of 6.5 days (IQR 5–8.3 days) and 7.5 days (IQR 6–9 days) after illness onset; Supplementary Table 2). H7N9 PCR-negative close contacts (herein referred to as H7N9-negative close contacts), H7N9 PCR-negative poultry workers (herein referred to as H7N9-negative poultry workers), outpatients with PCR-confirmed H1N1 or H3N2 infection (herein referred to as seasonal influenza outpatients) and hospitalized patients with PCR-confirmed H1N1 or H3N2 infection (herein referred to as seasonal influenza hospitalized).

### H7N9 infection in men is associated with reduced testosterone levels

To assess the role of sex hormones in the observed sex bias of H7N9 cases, we measured major sex hormones in all groups. In H7N9-infected men, circulating total testosterone levels were reduced compared to H7N9-negative poultry workers and H7N9-negative close contacts (18–49 yrs: $p < 0.0001$, $p < 0.0001$, ≥50 yrs: $p < 0.0001$, $p = 0.0117$) (Fig. 1a). Within the 18 to 49-year-old men, there was a trend towards a higher risk of death during the course of infection if total testosterone levels were low compared to those who survived ($p = 0.0399$) (Fig. 1b). We also measured the levels of free testosterone and sex hormone binding globulin (SHBG) to distinguish between SHBG-bound testosterone and unbound free testosterone. In H7N9-infected men of both age groups, free testosterone levels were below the lowest reference range according to the manufacturer's instructions (Supplementary Fig. 1a). SHBG levels, on the other hand, were within the normal reference range[14] in H7N9-infected men of both age groups (Supplementary Fig. 1b).

In H7N9-infected women, total testosterone levels were comparable to H7N9-negative close contacts and H7N9-negative poultry workers, albeit a slight tendency towards elevated testosterone levels was observed in those infected with H7N9 (Fig. 1c). This tendency of elevated testosterone levels was also observed in women who died during the course of H7N9 infection compared to those who survived (Fig. 1d). In H7N9-infected women, free testosterone levels were within the normal reference range, albeit slightly higher in those aged above 50 years who later died (Supplementary Fig. 1c). SHBG levels were within the normal reference range according to the manufacturer's instructions without relevant changes among the groups assessed (Supplementary Fig. 1d).

In men with seasonal H1N1 or H3N2 influenza infections, total testosterone levels were lower in those who were hospitalized than in those who were not hospitalized (outpatients), particularly among the 18- to 49-year-olds ($p = 0.0128$) (Fig. 1e). Free testosterone levels in men with seasonal influenza were distributed at the lowest percentile of the normal reference range in all groups assessed (Supplementary Fig. 1e). SHBG levels were within the normal reference range[14] in all groups (Supplementary Fig. 1f).

In women with seasonal influenza, there was a slight tendency towards elevated total testosterone levels in those who were hospitalized compared to outpatients (Fig. 1f). Free testosterone levels were

## Table 1 | Characteristics of study participants

| Age group | Study participants ($n = 369$) | Age (yrs) | | Sex (%) | |
|---|---|---|---|---|---|
| | | Median | Q1–Q3 | Male | Female |
| 18–49 yrs | H7N9 cases ($n = 44$) | 42 | 34–46 | 33 (75%) | 11 (25%) |
| | H7N9-negative close contacts ($n = 40$) | 35 | 27–41.3 | 29 (73%) | 11 (27%) |
| | H7N9-negative poultry workers ($n = 43$) | 41 | 36–46.5 | 32 (74%) | 11 (26%) |
| | Seasonal influenza outpatient ($n = 28$) | 26.5 | 23.3–32.8 | 16 (57%) | 12 (43%) |
| | Seasonal influenza hospitalized ($n = 15$) | 35 | 30–45 | 11(73%) | 4 (27%) |
| ≥50 yrs | H7N9 cases ($n = 54$) | 61 | 56–67 | 38 (70%) | 16 (30%) |
| | H7N9-negative close contacts ($n = 31$) | 62 | 54–70.5 | 16 (52%) | 15 (48%) |
| | H7N9-negative poultry workers ($n = 65$) | 55 | 53–62 | 45 (69%) | 20 (31%) |
| | Seasonal influenza outpatient ($n = 25$) | 61 | 56–67 | 12 (48%) | 13 (52%) |
| | Seasonal influenza hospitalized ($n = 24$) | 72 | 63–77 | 14 (58%) | 10 (42%) |

Data are median (25th quartile–75th quartile, Q1–Q3) or $n$ (%). Subjects with available testosterone or estradiol levels were included in the final analysis.

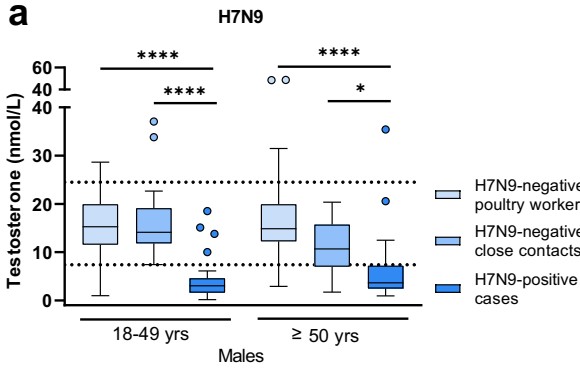

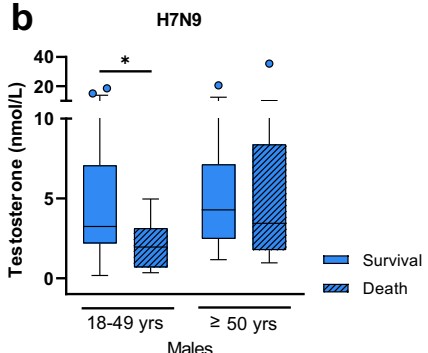

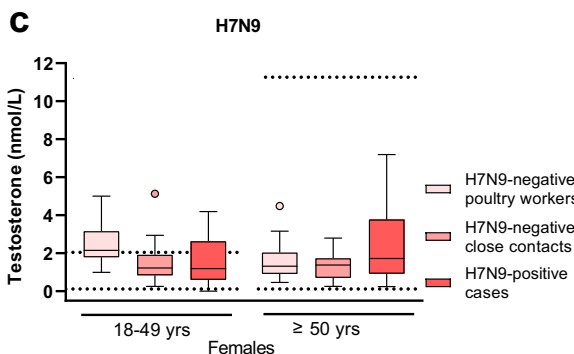

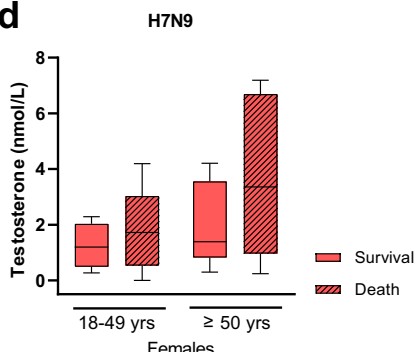

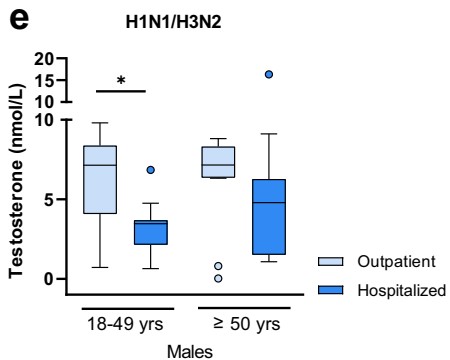

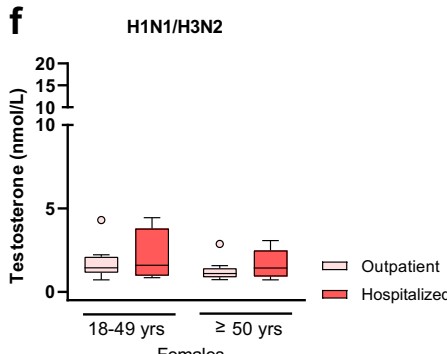

**Fig. 1 | Testosterone levels in H7N9-infected patients compared to control cohorts. a** Total testosterone expression levels were measured in sera from H7N9 IAV-infected male patients (18–49 yrs: $n = 33$, ≥50 yrs: $n = 38$) and compared to male controls as follows: (i) H7N9-negative poultry workers (18–49 yrs: $n = 32$, ≥50 yrs: $n = 45$) and (ii) H7N9-negative close contacts (18–49 yrs: $n = 29$, ≥50 yrs: n = 16). **b** Total testosterone expression levels in H7N9-infected male patients were compared in dependency of disease outcome (18–49 yrs survival/death: $n = 18/15$, ≥50 yrs survival/death: $n = 27/11$). **c** Total testosterone expression levels were measured in sera from H7N9 IAV-infected female patients (18–49 yrs: $n = 10$, ≥50 yrs: $n = 16$) and compared to female control groups as follows: (i) H7N9-negative poultry workers (18–49 yrs: n = 11, ≥50 yrs: $n = 20$) and (ii) H7N9-negative close contacts (18–49 yrs: $n = 11$, ≥50 yrs: $n = 15$). **d** Total testosterone expression levels in H7N9 IAV-infected female patients were compared in dependency of disease outcome (18–49 yrs survival/death: $n = 4/6$, ≥50 yrs survival/death: $n = 10/6$). **e, f** Total testosterone expression levels were measured in the plasma from

seasonal influenza-infected male (18–49 yrs outpatient/hospitalized: $n = 15/11$, ≥50 yrs outpatient/hospitalized: $n = 12/14$) (**e**) and female patients (18–49 yrs outpatient/hospitalized: $n = 12/4$, ≥50 yrs outpatient/hospitalized: $n = 13/10$) (**f**) and compared in dependency of disease severity (outpatient vs. hospitalized). **a, c** The age-specific reference range of testosterone in Chinese men and women was added as dotted lines in (**a**) and (**c**) (no age difference in testosterone in Chinese men)[14]. **a–f** Data are presented as Box-and-whisker plots (Tukey). The horizontal line in each box represents the median value. The 25th–75th percentiles represent the endpoints of the box. The whiskers stretch to the lowest and highest values within 1.5 times the interquartile range (IQR) from the 25th–75th percentiles. Dots represent outliers according to Tukey's definition. Unpaired, two-tailed non-parametric analysis (Mann–Whitney test) was used for comparisons between two groups (**b, d, e, f**). The Kruskal–Wallis test (**a, c**) was used for comparisons among the three groups. $P$ values were classified into four groups: *$p < 0.05$, **$p < 0.01$, ***$p < 0.001$, ****$p < 0.0001$. Source data are provided as a Source data file.

within the normal reference range in all women assessed, albeit higher in those above 50 years of age who were hospitalized compared to outpatients ($p = 0.0020$) (Supplementary Fig. 1g). SHBG levels were within the normal reference range in all women assessed (Supplementary Fig. 1h).

These data indicate that H7N9 infection in 18 to 49-year-old as well as in ≥50-year-old men, unlike in women, is associated with reduced total and free testosterone levels. Moreover, the data presented here support the hypothesis that low total testosterone levels in 18- to 49-year-old men are associated with fatal outcomes.

### H7N9 infection in men does not show an association with altered estradiol levels

Next, we measured estradiol levels, which represent another major sex hormone, in all cohorts. In H7N9-infected men, no alterations were observed among those infected with H7N9 or those who were H7N9-negative as close contacts or poultry workers. However, some H7N9-infected men presented elevated estradiol levels compared to H7N9-negative control cohorts (Fig. 2a). Similarly, some individuals who died during the course of H7N9 infection presented elevated estradiol levels compared to those who survived (Fig. 2b). In H7N9-infected women, estradiol levels were elevated in those above 50 years of age compared to H7N9-negative close contacts ($p = 0.0006$) (Fig. 2c). However, elevated estradiol levels showed no association with fatal disease outcome, albeit tendencies for poor outcomes in some women with elevated estradiol levels were observed (Fig. 2d).

In men with seasonal influenza infections, estradiol levels were not altered among those who were hospitalized or not hospitalized, albeit again, some individuals who were hospitalized presented elevated estradiol levels (Fig. 2e). In women infected with seasonal influenza, estradiol levels showed no change in those who were hospitalized compared to outpatients (Fig. 2f).

These data show no association between H7N9 infection in men and altered estradiol levels. In women, H7N9 infection is associated with elevated estradiol levels in those above 50 years of age. Since disease outcome in H7N9-infected women is not significantly affected, the role of estradiol in women requires further investigation.

### Elevated levels of inflammatory cytokines and chemokines involved in monocyte and granulocyte pathways are associated with fatal outcomes in H7N9-infected men

A complex cross-talk between sex hormones and cytokines with mutual regulatory activity is known to affect key immune pathways[15]. Thus, in order to elucidate potential interactions between sex hormones and innate immune responses, we first measured and described a panel of 29 different cytokines and chemokines in H7N9-infected men and women separately.

In both sexes, H7N9 infection was associated with high-level induction of cytokines and chemokines, in agreement with previous reports on cytokine storms observed in patients infected with avian influenza viruses[16,17] (Supplementary Fig. 2). However, some of these cytokines and chemokines were associated with fatal outcomes. In H7N9-infected men, elevated levels of G-CSF (18–49 yrs: $p = 0.0364$), GM-CSF (≥50 yrs: $p = 0.0438$), IL-8 (18–49 yrs: $p = 0.0021$, ≥50 yrs: $p = 0.0381$), IL-10 (≥50 yrs: $p = 0.0300$), IL-15 (18–49 yrs: $p < 0.0001$, ≥50 yrs: $p = 0.0322$), MCP-1 (≥50 yrs: $p = 0.0116$) and TNF-α (18–49 yrs: $p = 0.0182$) were associated with later death (Fig. 3a–g, o–q). In H7N9-infected women, decreased levels of Eotaxin (18–49 yrs: $p = 0.0061$) and elevated levels of MIP-1ß (18–49 yrs: $p = 0.0121$, ≥50 yrs: $p = 0.0216$) and EGF (≥ 50 yrs: $p = 0.0405$) were associated with fatal outcomes (Fig. 3h–n, r–t).

In seasonal H1N1/H3N2-infected men, on the other hand, elevated levels of IL-8 (18–49 yrs: $p < 0.0001$, ≥50 yrs: $p = 0.0111$), MIP-1α (18–49 yrs: $p = 0.0346$), IL-7 (18–49 yrs: $p = 0.0338$), IL-15 (18–49 yrs:

$p = 0.0291$), EGF (18–49 yrs: $p = 0.0390$), VEGF (18–49 yrs: $p = 0.0145$) and IFN-α2 (≥50 yrs: $p = 0.0102$) were associated with hospitalization compared to outpatients (Fig. 4a–g, o, p). In women infected with seasonal H1N1/H3N2 influenza viruses, increased levels of IL-8 (18–49 yrs: $p = 0.0148$, ≥50 yrs: $p = 0.0042$) and decreased levels of IFN-γ (≥50 yrs: $p = 0.0202$) and MIP-1ß (≥50 yrs: $p = 0.0137$) were associated with hospitalization (Fig. 4h–n, q–r).

These findings show that more cytokines and chemokines are associated with poor prognosis in H7N9- and seasonal H1N1/H3N2 influenza-infected men than in women. Herein, elevated levels of G-CSF, GM-CSF, IL-10, MCP-1, TNF-α and particularly IL-15 were associated with death in H7N9-infected men, unlike females or males infected with seasonal influenza.

### Inflammatory markers such as IL-15 are negatively associated with testosterone levels in H7N9-infected men

Next, we sought to identify potential interactions between the identified H7N9 male-specific cytokines and chemokines and testosterone levels. Therefore, we applied a linear regression model for all 29 cytokines/chemokines and testosterone levels assessed on the logarithmic scale (Fig. 5). Herein, we detected a negative association between testosterone levels and IL-6 ($R^2 = 0.3672$), IL-10 ($R^2 = 0.2387$), IL-15 ($R^2 = 0.5049$), IL-1RA ($R^2 = 0.2506$), and IL-1α ($R^2 = 0.2564$) in H7N9-infected 18–49-year-old men (Fig. 5a–e) as well as IL-6 ($R^2 = 0.2355$) and IL-10 ($R^2 = 0.2966$) in H7N9-infected men 50 years and older (Fig. 5f, g). In seasonal H1N1/H3N2-infected men, on the other hand, only G-CSF levels were negatively associated with testosterone levels (Supplementary Fig. 3).

These data suggest that testosterone levels are negatively associated with the expression of key cytokines involved in antiviral immune responses, such as the inflammatory cytokine IL-15, which can explain approximately 50% of the variation ($R^2 = 0.5049$) in testosterone levels in H7N9-infected males.

### In male mice, respiratory H7N9 influenza virus infection spreads to the testes and causes a reduction in circulating testosterone levels

To address whether the reduction in circulating testosterone levels detected in H7N9-infected men presents a causal relationship, we additionally used a murine model previously shown to provide a valuable preclinical model to study sex hormones in infection[11–13]. Therefore, male mice were infected with avian H7N9 influenza virus or treated with PBS or poly(I:C) as a control (Supplementary Fig. 4a). H7N9 virus was detected in the lungs as well as in the testes of infected mice on days 1, 3 and 6 p.i. (Fig. 6a, b) in line with respiratory inflammation (Supplementary Fig. S5). Pre-infection testosterone levels in the plasma of poly(I:C)-treated mice were considerably reduced on days 1 ($p = 0.0120$) and 3 p.i. ($p = 0.0469$) but not 6 p.i. (Fig. 6c–e). Similarly, H7N9 influenza virus infection reduced circulating testosterone levels compared to those before infection (1d.p.i: $p = 0.0137$, 3 d.p.i: $p = 0.0244$) (Fig. 6f–h). As a control, we measured estradiol levels which were not significantly affected except on day 1 or day 6 after infection ($p = 0.0063$, $p = 0.0019$) and day 1 after poly(I:C) treatment ($p = 0.0005$) (Supplementary Fig. S4j–o). Testes of H7N9-infected mice presented elevated pro-inflammatory cytokine and chemokine levels, such as MCP-1 (PBS or poly (I:C) vs. H7N9: $p = 0.0071$, $p = 0.0080$), IP-10 (PBS vs. H7N9: $p = 0.0472$), IL-10 (PBS or poly (I:C) vs. H7N9: $p < 0.0001$, $p = 0.0014$), MIP-1ß (PBS or poly (I:C) vs. H7N9: $p < 0.0001$, $p = 0.0004$) and IL-1ß (PBS or poly (I:C) vs. H7N9: $p < 0.0001$, $p = 0.0079$), compared to PBS- or poly(I:C)-treated animals at days 3p.i., the time point of highest viral replication (Fig. 6i–m; Supplementary Fig. 4b–i). Despite the elevated testicular cytokine response in H7N9-infected male mice, no alterations were observed on the histological level in the testes (Fig. 6n–p; Supplementary Fig. 4p–u).

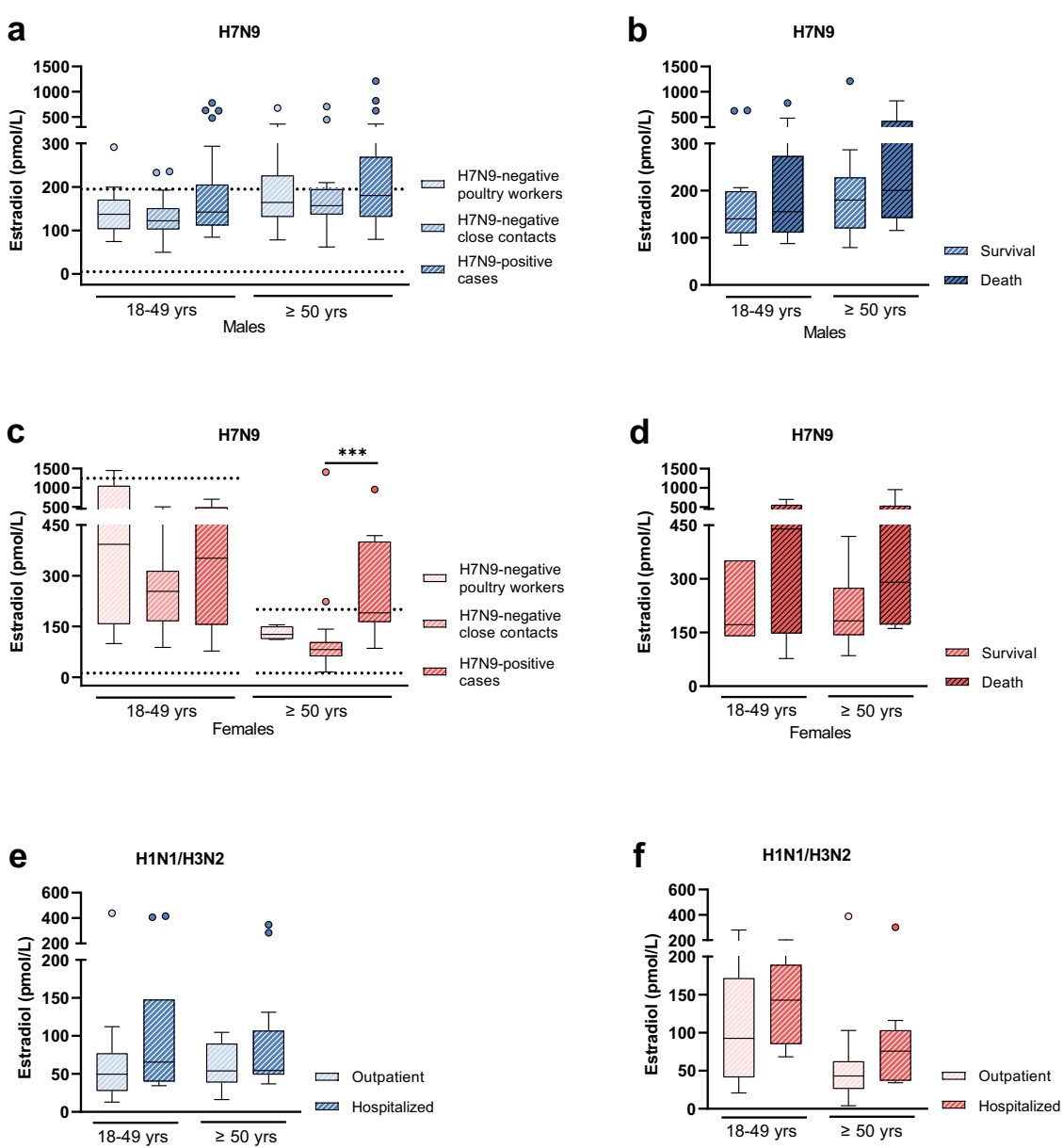

**Fig. 2 | Estradiol levels in H7N9-infected patients compared to control cohorts.** **a** Estradiol levels were measured in sera from H7N9 IAV-infected males (18–49 yrs: $n = 32$, ≥50 yrs: $n = 27$) and compared to male control groups as follows: (i) H7N9-negative poultry workers (18–49 yrs: $n = 15$, ≥50 yrs: $n = 19$) and (ii) H7N9-negative close contacts (18–49 yrs: $n = 29$, ≥50 yrs: $n = 16$); **b** Estradiol levels in H7N9 IAV-infected male patients were compared in dependency of disease outcome (18–49 yrs survival/death: $n = 18/14$, ≥50 yrs survival/death: $n = 17/10$); **c** Estradiol levels were measured in sera from H7N9 IAV-infected females (18–49 yrs: $n = 9$, ≥50 yrs: $n = 12$) and compared to female control groups as follows: i) H7N9-negative poultry workers (18–49 yrs: $n = 8$, ≥50 yrs: $n = 4$) and ii) H7N9-negative close contacts (18–49 yrs: $n = 11$, ≥50 yrs: $n = 15$). **d** Estradiol levels in H7N9 IAV-infected female patients were compared in dependency on disease outcome (18–49 yrs survival/death: $n = 3/6$, ≥50 yrs survival/death: $n = 6/6$). **e**, **f** Estradiol levels were measured in plasma from seasonal influenza-infected male (18–49 yrs outpatient/hospitalized: $n = 16/11$, ≥50 yrs outpatient/hospitalized: $n = 12/14$) (**e**) and female patients (18–49 yrs outpatient/hospitalized: $n = 12/4$, ≥50 yrs outpatient/hospitalized: $n = 13/10$) (**f**) and are compared in dependency of disease outcome (outpatient vs. hospitalization). The age-specific reference range of estradiol in Chinese men and women was added as dotted lines in (**a**) and (**c**) (no age difference in estradiol in Chinese men)[39]. **a–f** Data are presented as Box-and-whisker plots (Tukey). The horizontal line in each box represents the median value. The 25th–75th percentiles represent the endpoints of the box. The whiskers stretch to the lowest and highest values within 1.5 times the interquartile range (IQR) from the 25th–75th percentiles. Dots represent outliers according to Tukey's definition. Unpaired, two-tailed non-parametric analysis (Mann–Whitney test) was used for comparisons between two groups. The Kruskal–Wallis test was used for comparisons among the three groups. $P$ values were classified into four groups: *$p < 0.05$, **$p < 0.01$, ***$p < 0.001$, ****$p < 0.0001$. Source data are provided as a Source data file.

These data suggest that circulating testosterone levels are specifically reduced upon H7N9 virus infection but also upon poly(I:C) treatment of male mice. However, an inflammatory cytokine response was detected in the testes of H7N9-infected but not in those of poly(I:C)-treated mice.

## Discussion

Avian influenza A (H7N9) virus has repeatedly crossed species barriers and infected humans. Since its first emergence in 2013, H7N9 influenza has caused 1,568 laboratory-confirmed cases and 616 deaths, particularly in men[18,19]. Avian influenza viruses are considered as one of the

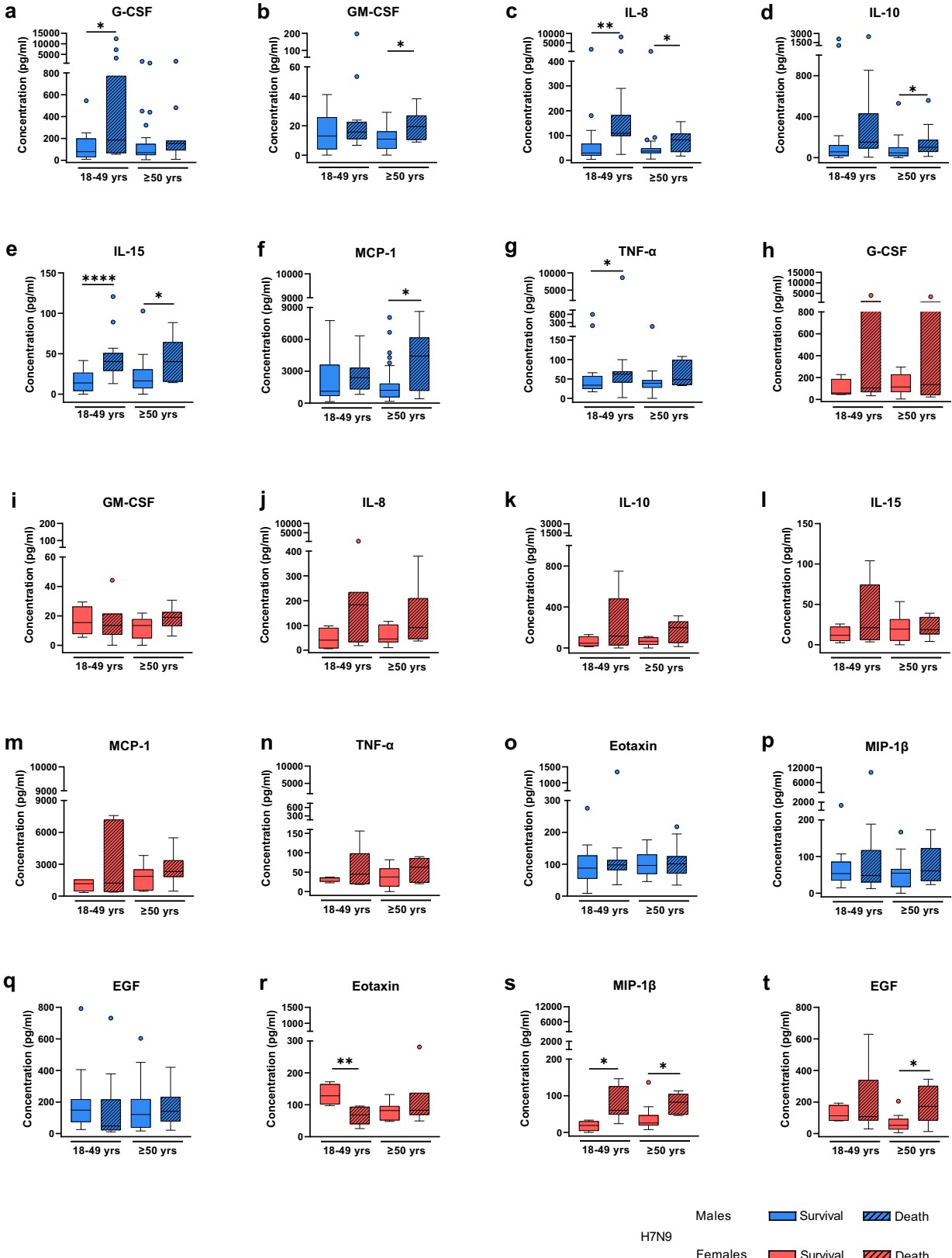

major candidates posing a high risk to cause the next pandemic. Although many aspects of avian H7N9 infection have been studied, a systematic analysis of the higher number of male H7N9 infections across five epidemic waves in China was still lacking.

In this study, we retrospectively analysed five groups (18–49 yrs and ≥50 yrs), including laboratory-confirmed H7N9 cases, influenza virus-negative H7N9 close contacts, epidemiologically linked poultry workers and outpatients and hospitalized patients with seasonal H1N1/H3N2 influenza infections. We systematically measured the levels of the main androgen (testosterone) and estrogen (estradiol) and explored their differences in the groups under study as well as their correlation with cytokine and chemokine responses

**Fig. 3 | Cytokine and chemokine expression profiles in H7N9-infected patients in dependency of disease outcome. a–t** Shown are the expression levels of cytokines and chemokines in the sera of H7N9 IAV-infected males (18–49 yrs survival/death: $n = 18/15$, ≥50 yrs survival/death: $n = 27/11$) and female patients (18–49 yrs survival/death: $n = 4/7$, ≥50 yrs survival/death: $n = 10/6$) who either survived or succumbed to the infection. The measurement was performed using a multiplex immunoassay. Cytokines/chemokines considerably dysregulated in males include: G-CSF (granulocyte colony-stimulating factor; **a**), GM-CSF (granulocyte-macrophage colony-stimulating factor; **b**), IL-8 (interleukin 8; **c**), IL-10 (interleukin 10; **d**), IL-15 (interleukin 15; **e**), MCP-1/CCL2 (monocyte chemoattractant protein 1; **f**), and TNF-α (tumour necrosis factor alpha; **g**). The same cytokines and chemokines were not altered in female H7N9 patients between both groups (**h**–**n**). Cytokines/ chemokines considerably dysregulated in females include: Eotaxin (**r**), MIP-1β/ CCL4 (macrophage inflammatory protein 1 beta; **s**); and EGF (epidermal growth factor; **t**). These cytokines were not significantly altered in males between both groups (**o**–**q**). Data are presented as Box-and-whisker plots (Tukey). The horizontal line in each box represents the median value. The 25th–75th percentiles represent the endpoints of the box. The whiskers stretch to the lowest and highest values within 1.5 times the interquartile range (IQR) from the 25th–75th percentiles. Dots represent outliers according to Tukey's definition. Unpaired, two-tailed non-parametric analysis (Mann–Whitney test) was used for comparisons between two groups. The Kruskal–Wallis test was used for comparisons among the three groups. $P$ values were classified into four groups: $*p < 0.05$, $**p < 0.01$, $***p < 0.001$, $****p < 0.0001$. Source data are provided as a Source data file.

known to be major drivers of influenza disease severity in both sexes.

This retrospective study has some limitations. By nature of the study, these data present secondary data use from the health care sector. Therefore, a systematic explorative statistical analysis was performed as a first translational step to generate hypotheses to be further explored in animal models later. Multiple adjustments with other factors influencing testosterone and estrogen estradiol were therefore not taken into account. Our data were obtained from the Influenza Surveillance Network in the local Center for Disease Control and Prevention (CDC), China, which is responsible for identifying and reporting all novel influenza cases nationwide. However, as the study design includes voluntary participation, it is never free from any selection bias. However, this effect may be ignored, because we do not have any information on underlying comorbidities present in the enrolled sample population that might have additionally affected sex hormone levels. It was reported that some metabolic comorbidities, such as obesity and/or type II diabetes, are also associated with changes in sex hormones irrespective of infection[20,21]. Therefore, the statistical data exploration is not free from confounding bias. The overall low sample size, particularly in female patients, limits the conclusions as well due to the low statistical power.

Thus, the data and results presented here may be interpreted as a first translational step. Future investigations are required in larger and sex-balanced groups to study the impact of sex hormones on influenza disease outcome in a structural epidemiological setting, taking additional individual data into account. In addition, sex hormone levels were measured at one time point. Future studies should monitor sex hormone levels during the course of infection to investigate their impact as biomarkers using a prospective setting.

Besides these limitations, the data presented here support a series of interesting hypotheses. Within this explorative process, we observed a strong suppression of total and bioavailable free testosterone levels in H7N9-infected men between 18–49 years and 50 and above years of age. Likewise, seasonal H1N1/H3N2 influenza-infected men with suppressed testosterone levels were more likely to be hospitalized than outpatients. However, the level of testosterone depletion was more severe in men with avian H7N9 influenza at both age groups than in men with seasonal influenza when referring to their median levels of total and free testosterone. Reduction of total testosterone levels in H7N9-infected men aged 18–49 years was further associated with lethal disease outcome, suggesting a prominent role of testosterone, particularly in avian H7N9 influenza-infected men.

Second, we hypothesize from our data that cytokines and chemokines are negatively associated with testosterone levels. IL-6 was negatively associated with testosterone levels in H7N9-infected male patients in both age groups. This is particularly interesting, since high IL-6 levels were repeatedly reported to be a poor prognostic marker for severe infections, including coronavirus disease 2019 (COVID-19)[22]. Importantly, the pro-inflammatory cytokine IL-15, which has been shown to be upregulated in severe influenza infection[23,24], showed the strongest association with testosterone in our data. Moreover, IL-15 is considered an ideal molecular adjuvant in universal influenza vaccines, which show robust cross-clade protection against H5N1[25] and heterosubtypic immunity against H7N9 infection, seasonal influenza and highly pathogenic H7N7 influenza A viruses[26]. Furthermore, IL-15 is also discussed as a component in immunotherapy in COVID-19 patients[27]. Among all the H7N9 infection-specific cytokines, we identified IL-10 to be additionally negatively associated with testosterone levels in H7N9-infected men. IL-10 is a master regulator of immunity to infection[28]. One of its main functions is to promote an anti-inflammatory environment. According to the data generated herein, low testosterone levels associated with high inflammatory cytokine levels, such as pro-inflammatory cytokines IL-6 and IL-15, are paralleled by elevated anti-inflammatory IL-10 levels, likely in response to the cytokine storm induced by H7N9 in men. These findings are in line with reports on the immunosuppressive function of testosterone[29–32].

Third, we used a murine model to address the question of whether testosterone levels in H7N9-infected men were reduced upon virus infection or due to other underlying factors. Therefore, we infected male mice and detected that respiratory H7N9 infection causes a significant reduction in circulating testosterone levels compared to levels before infection. Reduction of circulating testosterone levels was also observed upon poly(I:C) treatment of male mice. However, testicular inflammation measured as an increase in inflammatory cytokine and chemokine responses was only observed upon H7N9 infection, in line with the high virus load detected in the testes of infected mice. These data suggest that testosterone production may be repressed either directly through local testicular infection and inflammation or through systemic inflammation and its action on the hypothalamic–pituitary–gonadal axis[33–36].

In light of the current COVID-19 pandemic, which also highlights the male sex as a risk factor for hospitalization and death[37], our H7N9 study here might serve as a blueprint for future in-depth investigations. This knowledge may provide a basis for future individualized risk assessment in patients with emerging viral infections.

## Methods
### Ethics statement
**Humans.** Sampling from laboratory-confirmed H7N9 avian influenza cases, seasonal influenza cases, close contacts of H7N9 cases and poultry workers were reviewed and approved by the Ethics Committee of the National Institute for Viral Disease Control and Prevention, China CDC. Informed consent was obtained from all subjects in this study.

**Animals.** All animal experiments were performed in strict accordance with the German animal protection law (Behörde für Gesundheit und Verbraucherschutz, Hamburg, Germany; licensing number: N124/2021).

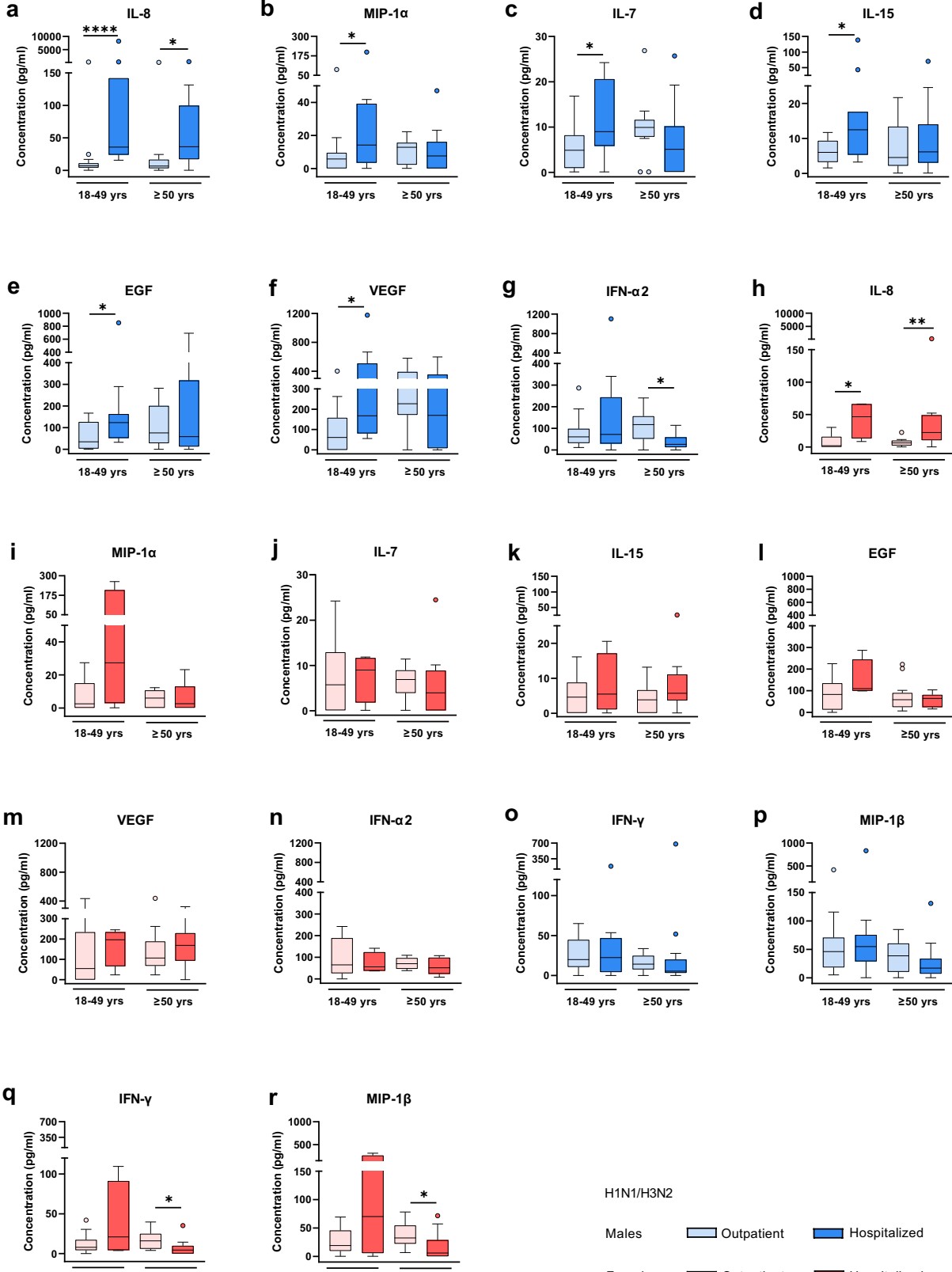

## Study participants

Stored serum samples from the acute phase of laboratory-confirmed H7N9 hospitalized patients were collected between 2014 and 2017 through the influenza surveillance network in the local Center for Disease Control and Prevention (CDC), China. H7N9 patients ($n = 98$) aged 18 years or older with complete epidemiological information (age, sex, illness onset date, sample collection date, outcome and antiviral treatment) and the required amount of serum were enrolled in this study. Once a patient was confirmed to have H7N9 influenza virus, serum samples and throat swabs from close contacts and epidemiologically linked poultry workers were collected through the influenza surveillance network of the local CDC, China. Swabs were

**Fig. 4 | Cytokine and chemokine expression profiles in seasonal H1N1/H3N2 influenza patients in dependency of disease outcome. a–r** Shown are the expression levels of cytokines and chemokines in the plasma from seasonal influenza outpatients and hospitalized patients, each divided into two age groups: 18–49 yrs (male outpatient/hospitalized: $n = 16/11$, female outpatient/hospitalized: $n = 12/4$) and ≥50 yrs (male outpatient/hospitalized: $n = 12/14$, female outpatient/hospitalized: $n = 13/10$). The measurement was carried out using a multiplex immunoassay. Cytokines/chemokines considerably dysregulated in either or both sexes include: IL-8 (interleukin 8; **a**, **h**), MIP-1α/CCL3 (macrophage inflammatory protein 1 alpha; **b**, **i**), IL-7 (interleukin 7; **c**, **j**), IL-15 (interleukin 15; **d**, **k**), EGF (epidermal growth factor; **e**, **l**), VEGF (vascular epidermal growth factor; **f**, **m**), IFN-α2 (interferon alpha 2; **g**, **n**), IFN-γ (interferon gamma; **o**, **q**), and MIP-1β/CCL4 (macrophage inflammatory protein 1 beta; **p**, **r**). Data are presented as Box-and-whisker plots (Tukey). The horizontal line in each box represents the median value. The 25th–75th percentiles represent the endpoints of the box. The whiskers stretch to the lowest and highest values within 1.5 times the interquartile range (IQR) from the 25th–75th percentiles. Dots represent outliers according to Tukey's definition. Unpaired, two-tailed non-parametric analysis (Mann–Whitney test) was used for comparisons between two groups. The Kruskal–Wallis test was used for comparisons among the three groups. $P$ values were classified into four groups: $*p < 0.05$, $**p < 0.01$, $***p < 0.001$, $****p < 0.0001$. Source data are provided as a Source data file.

first screened using real-time RT–PCR to identify influenza A and B viruses in the local CDC or the Chinese National Influenza Center. Samples positive for influenza A virus were further tested for influenza A (H7N9) virus. No influenza A (H7N9) virus was detected in close contacts or poultry workers. A total of 71 close contacts and 108 poultry workers were included in this study as control groups by matching age or sex of H7N9 cases. Archived plasma samples (EDTA plasma) from laboratory-confirmed outpatients ($n = 53$) and plasma samples of hospitalized patients ($n = 39$) infected with influenza A (H1N1) pandemic 2009 or influenza A (H3N2) viruses included in this study were selected based on the same age criteria. Plasma samples and throat swabs were collected simultaneously when the patient went to outpatient care or was hospitalized during the 2015–2016 influenza season in China. Plasma samples were archived when seasonal influenza virus was detected using real-time RT–PCR. All samples from study participants were stored at −80 °C until further analysis.

### Measurement of sex hormones in human samples
In human samples, the total concentrations of estradiol (17β-estradiol) and testosterone were measured in the clinical laboratory of the Sino-Japan Friendship Hospital (Beijing, China) from serum samples of study participants (H7N9 patients, H7N9 close contacts and poultry workers) using the Beckman Coulter DxI 800 Access Immunoassay System. The system uses the Access Estradiol Assay (REF 33540, BECKMAN COULTER) and Access Testosterone Assay (REF 33560, BECKMAN COULTER), which apply paramagnetic particles in a chemiluminescent immunoassay for the quantitative determination of total estradiol and testosterone levels (bound and unbound). For plasma (EDTA-plasma) samples from seasonal influenza-infected patients, the levels of estradiol and testosterone were measured using Abcam's 17β-Estradiol (ab 108667) and Testosterone (ab 174569) ELISA (Enzyme-Linked Immunosorbent Assay) kits, which have no cross-reaction with EDTA. The majority of circulating testosterone is bound to the sex hormone binding globulin (SHBG), but it also exists loosely bound to albumin and in the free state. The levels of SHBG and free testosterone were also measured from all H7N9 and seasonal influenza patients using Abcam's SHBG (ab 260070) and free testosterone (ab 178663) ELISA kits. Converting of testosterone, free testosterone and estradiol units to SI units (nmol/L or pmol/L) was performed either automatically from the measurement platform or manually according to the following formulas: for conversion of testosterone values given in ng/mL or free testosterone given in pg/mL, values were multiplied by the Factor 3.467; for conversion of estradiol values given in pg/mL, values were multiplied by the Factor 3.671 according to the manufacturer´s instructions.

### Animal experiments
All animal experiments were conducted at the Leibniz Institute for Virology (LIV), Hamburg, Germany, according to the German Animal Welfare Regulations. Male C57BL/6J mice (10 weeks old) were purchased from Envigo RMS Harlan Laboratories (Rossdorf, Germany). Mice were anaesthetized with $100 \, \text{mg kg}^{-1}$ ketamine and $10 \, \text{mg kg}^{-1}$ xylazine by intraperitoneal injection. The animals were intranasally inoculated with PBS, $5 \, \text{mg kg}^{-1}$ poly (I:C) or $10^3$ plaque forming units (p.f.u.) of H7N9 influenza A virus (A/Anhui/1/13). On day 1, 3 or 6 post infection (p.i.), respectively, the animals were anaesthetized with isoflurane, and blood was drawn by the retrobulbar route and collected in EDTA tubes. According to animal welfare protocols, blood can be drawn only twice per week from mice. Therefore, we divided the mice in the experiment into the following three independent groups: group #1 included pre-infection plasma and 1 day p.i. plasma from the same animal, group #2 included pre-infection plasma and 3 days p.i. plasma from the same animal, and group #3 included pre-infection plasma and 6 days p.i. plasma from the same animal. Blood samples were then centrifuged for 10 min at $2000 \times g$ and 4 °C. Lungs and testes of each time point post infection were collected and homogenized in PBS. All samples were stored at −80 °C before measurement. For histopathological examination of testes, samples were fixed in Bouin for 24 h, washed with 70% ethanol and subsequently embedded in paraffin using standard procedures. Sections were stained with haematoxylin and eosin. Body weight and mortality were monitored for 6 days.

### Measurement of sex hormones in murine samples
Animal sex hormones were measured at LIV, Hamburg, Germany. The total concentrations of estradiol (17β-estradiol) and testosterone were measured by using an enzyme immunoassay kit (DetectX, Arbor Assay) from diethyl ether-extracted plasma samples.

### Cytokine and chemokine measurement in human samples
Human cytokines/chemokines were measured at Sun Yat-sen University, Guangdong, China. Sera of H7N9 cases and plasma samples from seasonal influenza cases were analysed for a panel of 29 cytokines and chemokines (EGF, Eotaxin, G-CSF, GM-CSF, IFN-α2, IFN-γ, IL-10, IL12-P40, IL12-P70, IL-13, IL-15, IL-17A, IL-1RA, IL-1α, IL-1β, IL-2, IL-3, IL-4, IL-5, IL-6, IL-7, IL-8, IP-10, MCP-1, MIP-1α, MIP-1β, TNF-α, TNF-β and VEGF) by Luminex bead-based multiplex assay using a human cytokine/chemokine magnetic bead panel 96-well plate assay (HCYTOMAG-60K, Millipore, USA) according to the manufacturer's instructions. A smaller panel including 21 selected cytokines and chemokines showing significant differences in H7N9 cases (GM-CSF, IFN-α2, IFN-γ, IL-10, IL12-P40, IL12-P70, IL-13, IL-17A, IL-1RA, IL-1α, IL-1β, IL-5, IL-6, IL-7, IL-8, IP-10, MCP-1, MIP-1α, MIP-1β, TNF-α and TNF-β) was then used to measure the expression of these analytes for close contacts of H7N9 cases and poultry workers. For undetectable cytokines and chemokines (below the lowest detection limit) in all groups, we assigned a value of 0.1 pg/ml for analysis.

### Cytokine and chemokine measurement in murine samples
Murine cytokines/chemokines were measured at LIV, Hamburg, Germany. In mouse experiments, pulmonary cytokine and chemokine responses in the lung were analysed for Eotaxin, GM-CSF, IFN-γ, IL-1β, IL-2, IL-4, IL-5, IL-6, IL-10, MCP-1, MIP-1α, MIP-1β, TNF-α, IL-15 and VEGF by Bio-Plex Pro Mouse Cytokine kits (Bio-Rad). For undetectable cytokines and chemokines (below the lowest detection limit) in all groups, we assigned a value of 0.1 ng/ml for analysis.

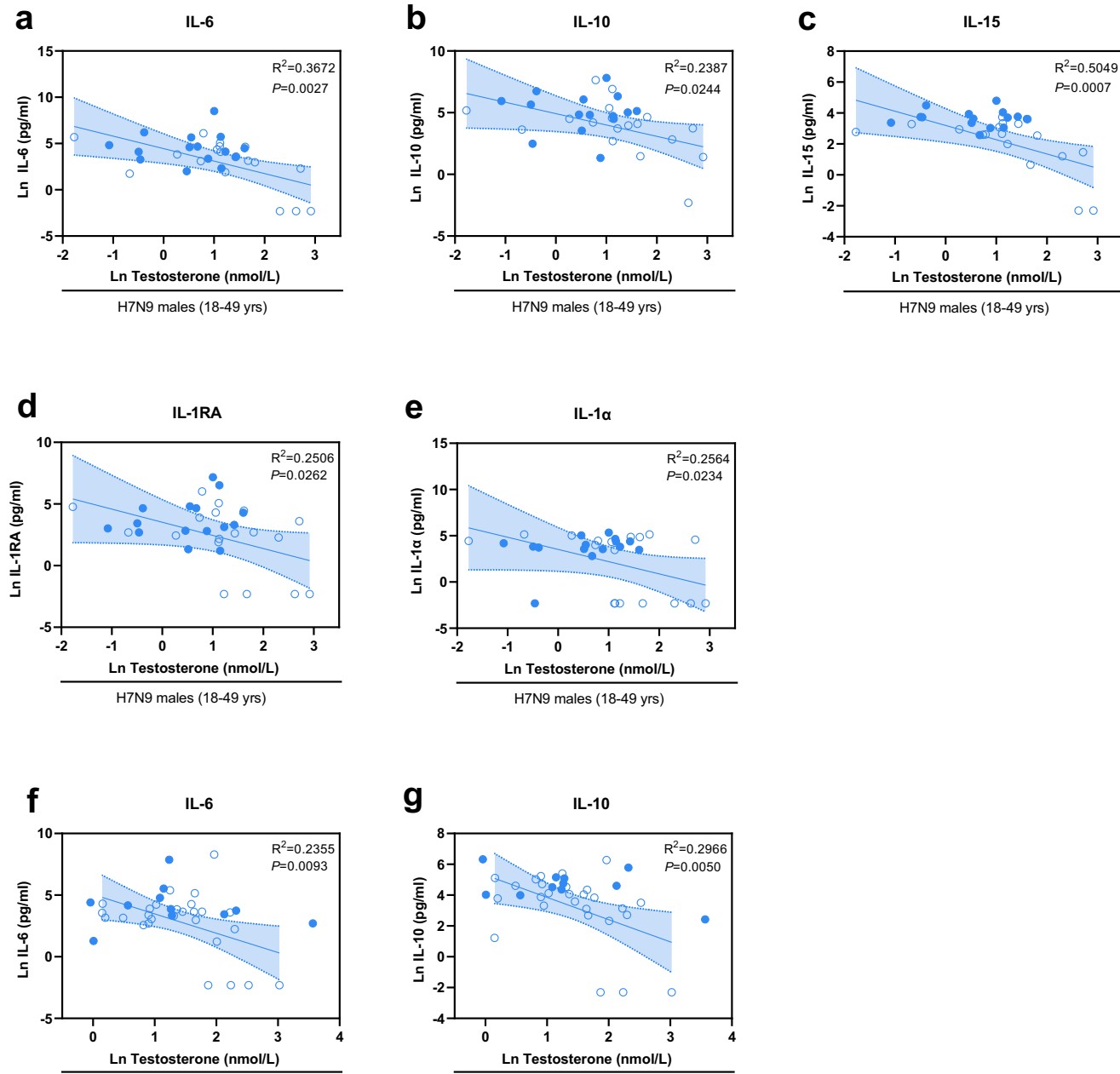

**Fig. 5 | Correlation of testosterone and cytokine/chemokine levels in men infected with H7N9 influenza. a–g** Shown is the observed regression between testosterone and cytokine/chemokine levels in H7N9 IAV-infected males. Natural logarithm transformed (Ln) testosterone expression levels measured in the sera from H7N9 males were plotted over the Ln transformed cytokine/chemokines levels. **a–e** Overall regression of testosterone with cytokines IL-6 (interleukin 6; **a**), IL-10 (interleukin 10; **b**), IL-15 (interleukin 15; **c**), IL-1RA (interleukin 1 receptor antagonist; **d**) and IL-1α (interleukin 1 alpha; **e**) observed from H7N9 younger males without interaction between deaths and survivors (18–49 yrs survival/death: $n = 18/15$); **f, g** Observed overall regression of testosterone with cytokines IL-6 (interleukin 6; **f**) and IL-10 (interleukin 10; **g**) in H7N9 older males without interaction between deaths and survivors (≥50 yrs survival/death: $n = 27/11$). **a–g** The best-fit line with 95% confidence intervals of the overall regression is shown in each figure. The centre of the error distribution is the regression line. Two-tailed linear regression was performed. Adjustment for multiple hypotheses was not necessary due to the explorative nature of the study. A hollow blue circle represents survival, and a solid blue circle represents death. R squared values are shown in the figures. Source data are provided as a Source data file.

## Cells and viruses

Madin-Darby Canine Kidney (MDCK; CCL-32) cells were obtained from ATCC and cultured in minimal essential medium (MEM) supplemented with 10% fetal bovine serum (FBS), 1% L-glutamine, and 1% penicillin-streptomycin at 37 °C and 5% $CO_2$. The cell line was regularly confirmed to be mycoplasma-negative by PCR. The influenza A virus used in this study is A/Anhui/1/13 (H7N9). Virus stocks were propagated and grown on MDCK cells, and virus titres were determined by plaque assay on MDCK cells following established protocols[38].

## Determination of viral titres in murine samples

All measurements were performed at LIV, Hamburg, Germany. Homogenization of organs was performed in 1 ml 1x PBS with 5 sterile, stainless-steel beads (∅ 2 mm, Retsch) at 30 Hz for 10 min in a MM400 mixer mill (Retsch). For determination of the viral titres, the tissue homogenates were titrated on MDCK cells in 10-fold serial dilutions for 30 min at 37 °C and 5% $CO_2$ and overlaid with MEM (Sigma–Aldrich) supplemented with 0.2% BSA, 1% L-glutamine, 1% penicillin–streptomycin, 1 µg ml⁻¹ L-1-tosylamido-2-phenylethyl chloromethyl ketone

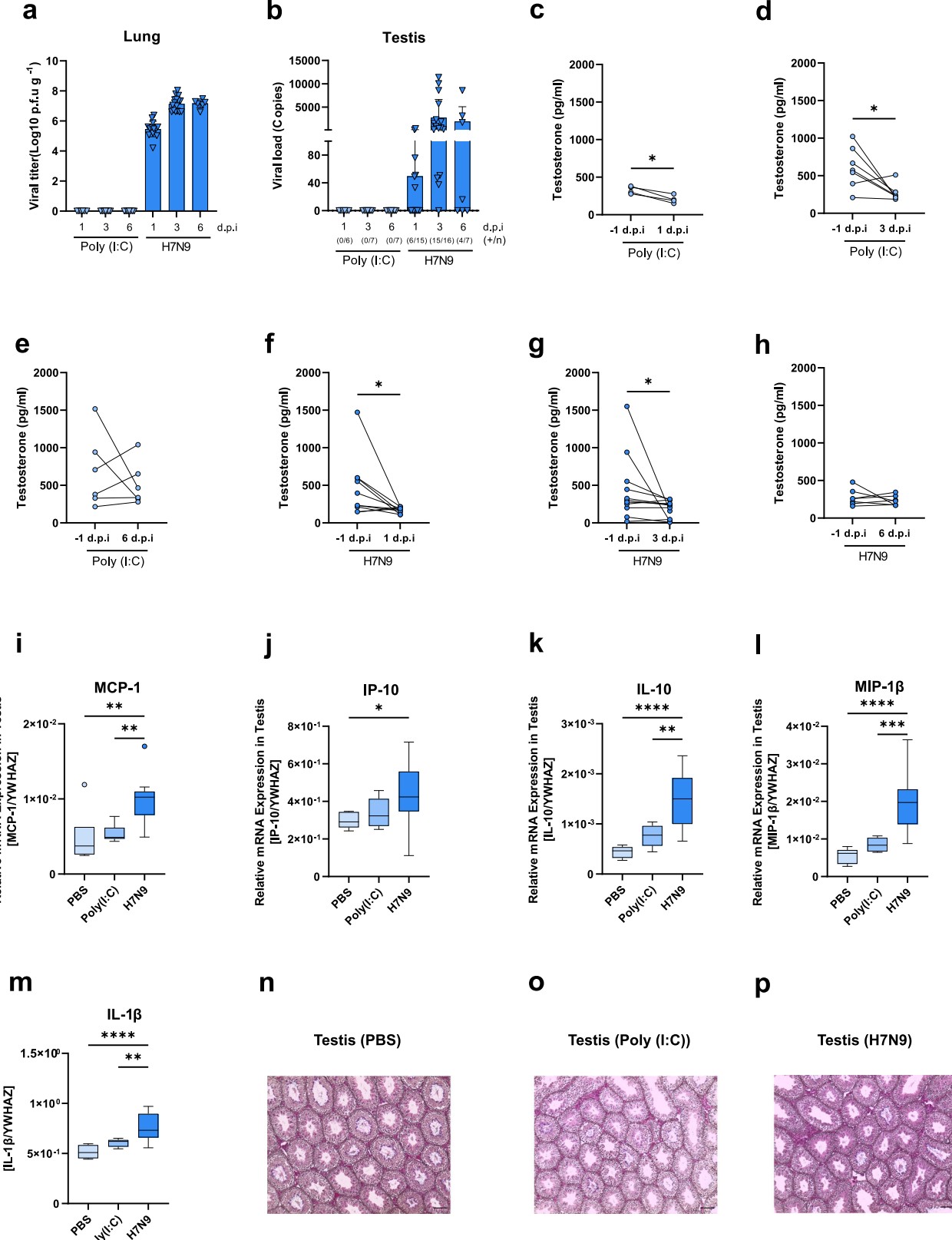

(TPCK)-treated trypsin (Sigma-Aldrich) and 1.25% Avicel. After 72 h p.i., the cells were washed with 1x PBS and fixed with 4% paraformaldehyde, and the plaques were visualized by crystal violet staining. Total RNA of testis was isolated from the aqueous phase after chloroform treatment using the innuPREP RNA Mini Kit 2.0 (Analytik Jena) according to the manufacturer's instructions with an additional DNase I treatment using

the RNase-free DNase Set (QIAGEN). cDNA was generated using random nonamer primers (Gene Link™, pd(N)9, 26-4000-06) and SuperScript™ III Reverse Transcriptase (Invitrogen) according to the manufacturer's instructions. For qPCR, specific primer pairs were used for the NP gene of A/Anhui/1/13 (H7N9). The reactions were set up in MicroAmp Optical 96-Well Reaction Plates (Invitrogen), including

**Fig. 6 | Infection with H7N9 reduces circulating testosterone levels and promotes an inflammatory response in the testes of male mice. a** Lung viral titres of mice control-treated with Poly (I:C) ($n = 6$, 7 and 7) or infected with H7N9 virus ($n = 15$, 16 and 7) at 1, 3 and 6 d.p.i. **b** Viral NP mRNA expression levels in the testes from mice control-treated with Poly (I:C) or infected with H7N9 virus at 1 d.p.i (Poly (I:C): $n = 6$, H7N9: $n = 15$), 3 d.p.i (Poly (I:C): $n = 7$, H7N9: $n = 16$) and 6 d.p.i. (Poly (I:C): $n = 7$, H7N9: $n = 7$). Number of mice with detectable H7N9 viral NP mRNA levels and the total number of mice are shown below the graph as (+/n). Data are presented as the mean values+SDs. **c–e** Before and post infection testosterone levels are shown from mice treated with Poly (I:C) ($n = 4$, 7 and 6) at 1, 3 and 6 d.p.i. **f–h** Before and post infection, testosterone levels were measured in mice infected with H7N9 virus ($n = 10$, 11 and 7) at 1, 3 and 6 d.p.i. **i–m** Significantly increased relative mRNA expression levels of inflammatory markers in the testes of H7N9-infected mice ($n = 16$) compared with PBS ($n = 7$)- or Poly (I:C) ($n = 7$)-treated mice at

3 d.p.i. Data are presented as Box-and-whisker plots (Tukey). The horizontal line in each box represents the median value. The 25th–75th percentiles represent the endpoints of the box. The whiskers stretch to the lowest and highest values within 1.5 times the interquartile range (IQR) from the 25th–75th percentiles. Dots represent outliers according to Tukey's definition. **n–p** Hematoxylin and eosin (HE)-stained paraffin sections from Bouin fixed tissue were evaluated via light microscopy. Shown are representative histological images of the testes of PBS ($n = 7$)-, or Poly (I:C) ($n = 7$)-control treated and H7N9-infected mice ($n = 7$) at 3 d.p.i. (10X). Scale bars (100 μm) are shown in the bottom right of each micrograph. **c–m** Unpaired or paired, two-tailed non-parametric analysis (Mann–Whitney test) or $t$ test was used for comparisons between two groups. One-way ANOVA or the Kruskal–Wallis test was used for comparisons among the three groups. $P$ values were classified into four groups: $*p < 0.05$, $**p < 0.01$, $***p < 0.001$, $****p < 0.0001$. Source data are provided as a Source data file.

Platinum SYBR Green qPCR SuperMix-UDG (Roche), forward and reverse primers and the cDNA template. RT–qPCR was conducted on a LightCycler 96 Real-Time PCR System (Roche). The standard curve was obtained by analysing serial dilutions of the *NheI*-linearized pHW2000-H7N9-NP vector construct. The following primer sequences were used for qRT–PCR:

NP forward: 5′-CCTGCTTGTGTGTACGGACT-3′
NP reverse: 5′-GGCTGTTTTGAAGCAGACGG-3′

## Determination of the mRNA expression levels of cytokines and chemokines in murine testes

All measurements were performed at LIV, Hamburg, Germany. Total RNA was isolated from the aqueous phase after chloroform treatment using the innuPREP RNA Mini Kit 2.0 (Analytik Jena) according to the manufacturer's instructions with an additional DNase I treatment using the RNase-free DNase Set (QIAGEN). cDNA synthesis was performed using random nonamer primers (Gene Link, pd(N)9, final concentration: 5 μM) and SuperScript III Reverse Transcriptase (Thermo Fisher Scientific) according to the manufacturer's instructions. Two microlitres of cDNA template was added to 10 μl FastStart Essential DNA Green Master Mix (Roche) and 300 nM forward and reverse primers. RT–qPCR runs were conducted on the LightCycler® 96 Real-Time PCR System (Roche) with endpoint fluorescence detection: 180 s at 95 °C and 45 amplification cycles (15 s at 95 °C, 10 s at 65 °C and 20 s at 72 °C). The Ct values of each sample were normalized to the Ct values of the reference gene *YWHAZ*, and the average ΔCt value ($n = 2$ technical replicates) was determined. The cytokines and chemokines used in this study are listed in Supplementary Table 1.

## Statistical analysis

A two-tailed non-parametric Mann–Whitney test or $t$ test was used for comparisons of two groups (paired or unpaired), and a non-parametric Kruskal–Wallis test or one-way ANOVA was used for comparisons of three groups. For linear regression analysis (two-tailed), values without normal distribution were transformed into natural logarithm (Ln) to describe quantitative relations. Due to the explorative nature of the study, no adjustment for multiple hypotheses was performed. Accordingly, statistical test p values were used as explorative measures at marked $*p < 0.05$, $**p < 0.01$, $***p < 0.001$, $****p < 0.0001$. Statistical analyses were conducted with GraphPad Prism 9.0.1 (GraphPad Software, Inc.), SAS®, version 9.4 TS level 1M5 (SAS Institute Inc., Cary, NC, United States) and IBM® SPSS® Statistics 27.

## Reporting summary

Further information on research design is available in the Nature Portfolio Reporting Summary linked to this article.

## Data availability

Source data are provided with this paper.

## Code availability

The code of SAS®, version 9.4 TS level 1M5 (SAS Institute Inc., Cary, NC, United States) is used including secured Macro-codes developed by the University of Veterinary Medicine Hannover (TiHo). Therefore, code is available for interested users by request only.

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

## Acknowledgements

This work was supported by the National Key Research and Development Program of China [2021YFC2300103 and 2016YFC1200200], the Guangdong Province Science and Technology Innovation Strategy Special Fund [2018A030310337], the Shenzhen Science and Technology Program [KQTD20180411143323605], the National Natural Science Foundation of China [81961128002], and the German Free and Hanseatic City of Hamburg [to LIV (G.G.)] and the German Federal Ministry of Health [to LIV (G.G.)]. We thank the Chinese National Influenza Surveillance Network and all staff at local CDCs in China for their contribution to sample collection and diagnosis for H7N9 and seasonal influenza A viruses. We would also like to thank the staff at Sino-Japan Friendship Hospital for the measurement of testosterone and estradiol. We also thank the study participants and their families. We thank Hanna Jania, Annette Gries, Anna Lüttjohann and Eva Wahle for their excellent technical support.

## Author contributions

G.G. and Y.L.S. conceived and designed the study. Y.L.S and D.Y.W. organized and coordinated the field investigations. S.B. and T.B. established sex hormone measurements, and Y.K.C., T.B. and T.T.J. conducted sex hormone measurements in human samples. Y.K.C. conducted cytokine and chemokine measurements and real-time RT–PCR for seasonal influenza viruses in human samples. J.D. performed real-time RT–PCR on close contacts of H7N9 patients and poultry workers. T.C., J.Y., L.J.W. and D.Y.W. provided and overviewed the epidemiological information from the study participants. S.B., T.B., S.S.B. and N.K.M. conducted animal experiments and performed measurements. A.M. performed histological analysis of the testes. L.K. and B.S. conducted statistical analyses, L.K., B.S. and A.Z. evaluated statistical analyses. T.B., S.B. and S.S.B. developed all figures. T.B. and G.G. wrote the manuscript. All authors revised the manuscript.

## Funding

## Competing interests

The authors declare no competing interests.
