## [Peer Review File · Nature Communications]

Reviewer comments, first round review

Reviewer #1 (Remarks to the Author):

In China in 2014-2017, ~70% of cases in an epidemic of avian influenza A (H7N9) occurred in men. Coupled with published reports showing influenza A infection reduces testosterone levels in mice, the gender disparity in human H7N9 cases led to the hypothesis that low testosterone was associated with poor outcome and high levels of serum pro-inflammatory cytokines. The manuscript uses a cohort of influenza H7N9 cases to determine the relationship between death/survival, testosterone and estradiol levels and serum cytokine levels in males and females. The H7N9 cases are compared to a cohort of individuals infected with seasonal influenza and to control cohorts of H7N9-negative close contacts and influenza-negative poultry workers. The lab assays appear to be straightforward, and the matching of individuals in the control and case cohorts seems acceptable. The data show that in H7N9-infected individuals, low testosterone levels were associated with death and high inflammatory cytokines, especially in 18-49 year old men. In contrast, cytokine levels did not correlate with testosterone in female cases.

This type of study is important – we know little about how sex hormone levels impact infectious disease outcome in humans. While some of the data are striking (testosterone levels are lower in H7N9 cases compared to control cohorts), other data show less robust significance, and conclusions seem driven by outliers. Minimal attempt was made to explain what disparate cytokine profiles may mean for disease outcome.

A number of concerns should be addressed.

1. It is stated that ~70% of H7N9 cases were males. A major risk factor was exposure at live poultry markets. The manuscript should state whether more men would have been likely to be present at these markets, as it seems possible that more cases were in men because more men were exposed to infected birds. It also seems possible that the men became H7N9 cases because they harbored comorbidities not present in control groups. It is not clear whether men were more likely to be H7N9 cases because their testosterone was low, or if the viral infection suppressed their testosterone levels.
2. Line 82 lacks a reference for the “we have recently shown...” statement.
3. Fig. 1 shows that testosterone levels were significantly lower in male H7N9 cases compared to the other cohorts – these data are striking. However, the correlation between testosterone levels and survival or death seems less significant in panel B. While a t test showed that testosterone levels were higher on average in survivors ($p < 0.05$), this statistic seems driven by 3 outliers. More individuals may be needed to make this point more convincing – in fact it seems that the graph does not contain $n=44$ for 18-49 yrs or $n=54$ for >50 yrs (stated cohort sizes).
4. Levels of cytokines induced by seasonal or H7N9 flu do not seem to differ much between males and females.
5. The plots showing correlations between testosterone levels and cytokine amounts do not seem to contain the entire cohort. The number of individuals analyzed should be reported. Again, the number of survivors with higher testosterone and driving the correlations seems low.
6. The study is more descriptive than mechanistic.

Reviewer #2 (Remarks to the Author):

In their manuscript, Chen et al have used a retrospective study design to investigate the

relationship between testosterone and disease outcomes in H7N9 infected men. They found that testosterone was lower in H7N9 infected men, with low testosterone in males associated with mortality and increased proinflammatory serum cytokine concentrations. Though a relationship between testosterone and experimental influenza A virus (IAV) outcomes has been established in murine models, this is the first time these findings have been extended to human cohorts. As such, these results are both novel and timely given the recent strong association between biological sex and COVID-19 outcomes. However, there are a few concerns with the underlying data from which these conclusions are drawn.

Major:

Total testosterone quantification is relatively simple to perform but higher serum concentrations of total testosterone may not necessarily equate to greater androgenic effects. Are there reasons why serum concentrations of free or bioavailable testosterone were not quantified in the current study? Could the authors quantify sex hormone binding globulin (SHBG) to gain a further understanding of the relationship between total testosterone and potential androgenic activity? As an example, SHBG concentrations increase with age in some men, leading to a faster age-related decline in free relative to total testosterone. This could influence the lack of an observed correlation between total testosterone and H7N9 related mortality in males aged 50 and over.

Cytokine/chemokine assays were run on serum samples for H7N9 infected patients and plasma samples for seasonal IAV infected patients. Given, the potential for sample type to influence the results of the Luminex magnetic bead system (doi: 10.1007/s12026-014-8491-6.), are these comparisons across groups valid? Could validation assays, or spiked controls, be run to increase confidence that the conclusions derived from these comparisons are not influenced by differences between serum and plasma? How was the plasma collected and processed (i.e. was EDTA or Sodium Heparin used)?

As with the cytokine/chemokine data, could the use of plasma from seasonal IAV infected patients and serum from H7N9 infected patients confound the between group differences in hormone concentrations? It should at least be considered as a possible explanation for the significant differences observed in Fig. 1C and Fig. 2A.

Minor:

It is curious that H7N9 infection in males is associated with low testosterone relative to the control groups while seasonal influenza infection is not. Could the authors speculate as to why this may be? This seems discordant with murine studies by the authors demonstrating testosterone suppression following H1N1 IAV infection. Is low testosterone an independent risk factor for hospitalization following IAV infection in males? Could the use of outpatients for the seasonal IAV cohort explain the lack of testosterone suppression in these patients?

Can the authors comment in the discussion on the possible reasons why low total testosterone is associated with H7N9 mortality in 18-49 year old males, but not those greater than or equal to 50 years of age? Do differences in total H7N9 case fatality rates occur between these two age groups?

Point-by-point response

Reviewer #1:

In China in 2014-2017, ~70% of cases in an epidemic of avian influenza A (H7N9) occurred in men. Coupled with published reports showing influenza A infection reduces testosterone levels in mice, the gender disparity in human H7N9 cases led to the hypothesis that low testosterone was associated with poor outcome and high levels of serum pro-inflammatory cytokines. The manuscript uses a cohort of influenza H7N9 cases to determine the relationship between death/survival, testosterone and estradiol levels and serum cytokine levels in males and females. The H7N9 cases are compared to a cohort of individuals infected with seasonal influenza and to control cohorts of H7N9-negative close contacts and influenza-negative poultry workers. The lab assays appear to be straightforward, and the matching of individuals in the control and case cohorts seems acceptable. The data show that in H7N9-infected individuals, low testosterone levels were associated with death and high inflammatory cytokines, especially in 18-49 year old men. In contrast, cytokine levels did not correlate with testosterone in female cases.

This type of study is important – we know little about how sex hormone levels impact infectious disease outcome in humans. While some of the data are striking (testosterone levels are lower in H7N9 cases compared to control cohorts), other data show less robust significance, and conclusions seem driven by outliers. Minimal attempt was made to explain what disparate cytokine profiles may mean for disease outcome. A number of concerns should be addressed.

We thank the reviewer for his/her valuable comments and criticism. We have now addressed all comments as described in detail below.

#1: It is stated that ~70% of H7N9 cases were males. A major risk factor was exposure at live poultry markets. The manuscript should state whether more men would have been likely to be present at these markets, as it seems possible that more cases were in men because more men were exposed to infected birds. It also seems possible that the men became H7N9 cases because they harbored comorbidities not present in control groups. It is not clear whether men were more likely to be H7N9 cases because their testosterone was low, or if the viral infection suppressed their testosterone levels.

This is a very important point. To our knowledge, there is no solid evidence that compared men to women who have a higher likelihood to be exposed to infected poultry in live poultry markets in China. Moreover, a modelling study suggested that the increased risk in older men is also not due to higher exposure time to live poultry market (Rivers, C, et al, 2013). We agree that men with existing comorbidities may have lower testosterone levels in general, however this information was not available for the cohorts presented in this study. We have now included this as a limitation of the retrospective setting of our study.

With respect to the causality of our findings, we have now addressed this experimentally. We now show that H7N9 replicates in Leydig cells of the testis and causes testosterone depletion in male mice (**see new Figure 6**) reflecting the observations in H7N9 infected men.

#2: Line 82 lacks a reference for the “we have recently shown...” statement.

This was amended.

#3: Fig. 1 shows that testosterone levels were significantly lower in male H7N9 cases compared to the other cohorts – these data are striking. However, the correlation between testosterone levels and survival or death seems less significant in panel B. While a *t* test showed that testosterone levels were higher on average in survivors ($p < 0.05$), this statistic seems driven by 3 outliers. More individuals may be needed to make this point more convincing – in fact it seems that the graph does not contain $n=44$ for 18-49 yrs or $n=54$ for >50 yrs (stated cohort sizes).

We have now included an expert in statistics who has scrutinized all raw data. Since all the values were not normally distributed, the comparison of testosterone levels between both groups (that is, survival vs death) was performed by non-parametric analysis (Mann-Whitney test) which is not affected by outliers. Besides, by referring to clinical range of testosterone in adult males (6.07-27.10 nmol/L) from the hospital, values lower than 6.07 nmol/L were defined as low testosterone levels and those higher or equal to 6.07 nmol/L as normal levels for a categorical variable analysis. Still, differences between deaths and survivals in males aged 18-49 years were statistically significant according to a *Chi-squared test* ($P=0.048$) (data not shown in this revision). Furthermore, we have now enrolled a novel cohort of hospitalized seasonal influenza cases. Similar to the H7N9 cohort, we observed a significant reduction in testosterone levels in hospitalized males aged 18-49 yrs compared to outpatients that were infected with seasonal influenza viruses (**revised Figure 1e**). These results further strengthen the association of reduced testosterone levels and influenza disease severity, particularly in male patients.

We further agree that including more samples into the H7N9 cohort would increase the power of our analysis. Nevertheless, according to our enrolling criteria, we could only include samples with complete epidemiological information and required amount of serum in this study. For better interpretation of the data presented in new Figure 1b, we have now listed the sample sizes (that is, 18-49 yrs survival/death: $n=18/15$, ≥ 50 yrs survival/death: $n=27/11$) in the revised figure legend. The number of $n=44$ (18-49 yrs) and $n=54$ (≥ 50 yrs) is the total number of H7N9 male and female patients.

#4: Levels of cytokines induced by seasonal or H7N9 flu do not seem to differ much between males and females.

This is correct for the raw data (**see Figure S2**). However, divided cytokine responses in H7N9 infected patients who survived infection and those who died highlights that more inflammatory cytokines are associated with poor disease outcome in infected men than women. Since there is a mutual complex regulation between sex hormones and cytokines (some cytokines have androgen binding elements in their promoters), we performed linear regression analysis to identify potential interactions between cytokines and sex hormones. In the **new Figure 5**, we show that key cytokines, such as IL-6 and IL-10 (which seems to be specific for H7N9 infection) are negatively associated with testosterone levels in H7N9 infected men. Since sex hormone levels have sex dependent reference values, our findings suggest that assessing cytokine and chemokine responses would probably make more sense to be analyzed in association with the respective sex hormone status.

#5: The plots showing correlations between testosterone levels and cytokine amounts do not seem to contain the entire cohort. The number of individuals analyzed should be reported. Again, the number of survivors with higher testosterone and driving the correlations seems low.

The number of individuals analyzed is now reported in the revised figure legends. In order to reduce deviation of the raw data, values of testosterone and cytokines were logarithm transformed in our linear regression analysis. Our primary goal is to understand the overall relationship between testosterone levels and cytokine expression in H7N9 infected male cases. According to the results, it is fair to conclude that several cytokines were negatively correlated with testosterone levels in H7N9 males of both age groups, such as IL-6 or IL-10 as described above (**see Figure 5**). The measurement of R^2 indicated about 23.55%-50.49% variance of testosterone levels can be explained by specific cytokines in this regression model. Although the R^2 is at low or medium level, the correlation of testosterone levels and cytokines can't be ignored.

#6: The study is more descriptive than mechanistic.

We completely agree with the reviewer's comment. In order to address this limitation, we performed new experiments. We analyzed the ability of H7N9 influenza virus to replicate in Leydig cells of the testis *ex vivo* and to modulate plasma testosterone levels *in vivo* in infected male mice. We now show that H7N9 influenza virus is able to replicate in rodent Leydig cell lines to higher levels than H1N1 or H3N2 influenza (**see new Figure 6a**). H7N9 replication in Leydig cells could be further confirmed in primary Leydig cells (**see new Figure 6b**). Finally, infection of male mice with H7N9 caused depletion in circulating testosterone levels (**see new Figure 6c**). These data indicate that H7N9 is the causative agent of testosterone depletion in males. Furthermore, these new data suggest that H7N9 replication in male gonadal cells might be the reason for impairment of testosterone synthesis. The fact that H7N9 replicates to higher levels in Leydig cells than H1N1 and H3N2 further confirms the observations in male patients where testosterone levels were more severely reduced in H7N9 infected men compared to men infected with seasonal influenza.

Reviewer #2:

In their manuscript, Chen et al have used a retrospective study design to investigate the relationship between testosterone and disease outcomes in H7N9 infected men. They found that testosterone was lower in H7N9 infected men, with low testosterone in males associated with mortality and increased proinflammatory serum cytokine concentrations. Though a relationship between testosterone and experimental influenza A virus (IAV) outcomes has been established in murine models, this is the first time these findings have been extended to human cohorts. As such, these results are both novel and timely given the recent strong association between biological sex and COVID-19 outcomes. However, there are a few concerns with the underlying data from which these conclusions are drawn.

We thank the reviewer for his/her valuable comments and criticism. We have now addressed all comments as described in detail below.

Major:

#1: Total testosterone quantification is relatively simple to perform but higher serum concentrations of total testosterone may not necessarily equate to greater androgenic effects. Are there reasons why serum concentrations of free or bioavailable testosterone were not quantified in the current study? Could the authors quantify sex hormone binding globulin (SHBG) to gain a further understanding of the relationship between total testosterone and potential androgenic activity? As an example, SHBG concentrations increase with age in some men, leading to a faster age-related decline in free relative to total testosterone. This could influence the lack of an observed correlation between total testosterone and H7N9 related mortality in males aged 50 and over.

We have now measured the levels of free testosterone and SHBG as suggested by the reviewer (**see new Figure S1**). Free testosterone levels are below reference range in H7N9 infected men reflecting the total testosterone levels. SHBG levels are within normal reference range in H7N9 infected men. This suggests that reduction in total testosterone levels is paralleled by reduction of free testosterone levels likely affecting biological activity. Since SHBG levels are within normal reference ranges, it is unlikely that age-dependent increase in SHBG accounts for low testosterone levels.

#2: Cytokine/chemokine assays were run on serum samples for H7N9 infected patients and plasma samples for seasonal IAV infected patients. Given, the potential for sample type to influence the results of the Luminex magnetic bead system (doi: 10.1007/s12026-014-8491-6.), are these comparisons across groups valid? Could validation assays, or spiked controls, be run to increase confidence that the conclusions derived from these comparisons are not influenced by differences between serum and plasma? How was the plasma collected and processed (i.e. was EDTA or Sodium Heparin used)?

We agree that cross-comparison of parameters measured in plasma and serum samples is difficult. In order to solve this discrepancy, we have now separated the cohorts into the H7N9 cohort where all parameters including of the H7N9-negative controls were measured in serum samples. In order to allow cross-comparison for the seasonal influenza cohorts, we have now recruited an additional seasonal influenza cohort of hospitalized patients where in

both cohorts now the parameters were measured in plasma samples. These **new data** are now shown in the accordingly **revised Figures 1 and 2**.

#3: As with the cytokine/chemokine data, could the use of plasma from seasonal IAV infected patients and serum from H7N9 infected patients confound the between group differences in hormone concentrations? It should at least be considered as a possible explanation for the significant differences observed in Fig. 1C and Fig. 2A.

We fully agree with this criticism. Therefore, we have now included new cohorts to compare serum samples only (H7N9 cohort and controls) and plasma samples only (seasonal influenza cohorts). See **revised Figures 1 and 2** as well as response to comment#2.

Minor:

#1: It is curious that H7N9 infection in males is associated with low testosterone relative to the control groups while seasonal influenza infection is not. Could the authors speculate as to why this may be? This seems discordant with murine studies by the authors demonstrating testosterone suppression following H1N1 IAV infection. Is low testosterone an independent risk factor for hospitalization following IAV infection in males? Could the use of outpatients for the seasonal IAV cohort explain the lack of testosterone suppression in these patients?

This discrepancy is now better presented including the new seasonal influenza cohort as suggested by the reviewer. In both, H7N9 and H1N1/H3N2 infected younger male patients, reduced testosterone levels are associated with poor prognosis (death in H7N9 and hospitalization in H1N1/H3N2). However, the degree of testosterone reduction is more severe in H7N9 infected men compared to H1N1/H3N2 infected men (**new Figure 1b,e and Supplementary Figure 1 a,e**). This is reflected by the ability of H7N9 to replicate more efficiently in Leydig cells compared to H1N1 or H3N2 influenza virus (**new Figure 6a**). We have now discussed this potential causative link between the ability to replicate in Leydig cells and circulating testosterone levels.

#2: Can the authors comment in the discussion on the possible reasons why low total testosterone is associated with H7N9 mortality in 18-49 year old males, but not those greater than or equal to 50 years of age? Do differences in total H7N9 case fatality rates occur between these two age groups?

We agree with this seeming discrepancy. One possible confounder might be the increasing presence of comorbidities in the elderly (other than metabolic comorbidities that could additionally affect the hormone status). Thus a higher sample size as well as knowledge on existing comorbidities in the elderly might be more conclusive and should be addressed in a future prospective setting. This limitation is now discussed in the discussion.

Reviewer comments, second round review

Reviewer #1 (Remarks to the Author):

I find the revised manuscript to be significantly improved. I have no additional concerns.

Reviewer #2 (Remarks to the Author):

This manuscript is a resubmission by Chen et al that primarily uses a retrospective study design to explore the role sex hormones play in explaining the male bias in H7N9 influenza infection. The authors find that low testosterone in males is associated with increased severity of H7N9 and seasonal (i.e., H1N1 and H3N2) influenza infection, with higher serum concentrations of proinflammatory cytokines and chemokines also being associated with low testosterone and severe H7N9 infection. Estrogen concentrations were not associated with influenza infection severity in males.

Overall, this is a well-organized study that strongly supports an association between low testosterone and increased influenza infection severity. This is a novel and significant finding in human patients that has previously only been shown with animal models of influenza infection. Moreover, the authors have now adequately addressed my concerns about cross-comparisons using different sample types and the added hospitalized seasonal influenza cohort greatly strengthens the relationship between low testosterone in males and severe outcomes with influenza. However, the addition of Leydig cell infection data, while intriguing, is unconvincing and inadequate evidence is presented to support the conclusion that it is the direct infection of these cells that results in lower circulating testosterone. As a result, there are several questions and concerns that must be addressed if this data is to be included the current manuscript. Alternately, the Leydig cell data may be better saved for a future manuscript where the hypothesis can be fully developed and explored.

#1: The evidence presented in support of influenza A virus infection of Leydig cells is rather unconvincing. Virus recovery is several logs lower than what would be expected with a relatively high MOI infection of known permissive/susceptible cell types (e.g., MDCK and primary nasal epithelial cells) and is declining or is stagnate over time. Additionally, several controls and the limit of detection are missing from these assays. The inclusion of a back-titer of the inoculation, an eclipse phase time-point (i.e., at 1 or 2hrs), and a 12hr time-point would help convince the readers that the relatively low levels of virus recovered at later times points are the result of a productive viral infection and not just the remnants of the inoculation process. Moreover, demonstrating viral receptor expression by Leydig cells and productive viral replication through a second means, such as the production of cell-associated viral proteins, would also strengthen this argument.

#2: The authors broadly conclude that it is the severe H7N9 infection that results in decreased male testosterone concentrations. However, the opposite relationship has also been demonstrated in murine models of influenza A virus infection (IAV) with low testosterone at the time of infection predisposing male mice to more severe outcomes with IAV infection. Given that both directional relationships may be true with H7N9 infection, and the lack of pre-infection testosterone concentration data, it seems as if only the conclusion that there is an association between low testosterone and increased severity H7N9 and seasonal influenza infection can be made in the current study. Likewise, were pre-infection testosterone concentrations measured for the mice in figure 6C? Given the high variability in testosterone concentrations, showing within mouse declines in testosterone following H7N9 infection would be more convincing than just showing a difference between the two treatment groups.

#3: Were cytopathic effects seen with the infection of the LC540 and primary Leydig cells? Was evidence of testicular inflammation or other pathological findings present in study participants or experimentally infected mice? If available, the inclusion of this data would greatly enhance the

argument that it is the direct infection of these cells that results in the decreased systemic testosterone concentrations as opposed to the indirect effects inflammatory mediators have on the hypothalamic-pituitary-gonadal axis (doi.org/10.1210/jc.2015-3611 & doi.org/10.1089/107999099312948) demonstrated by others.

#4: Several of the box and whisker plots are missing one or both of their whiskers. This may just be an unfortunate artifact of the gaps added to the axes of these figures.

#5: Is it possible to include the reference ranges for total testosterone and estradiol in figures 1 and 2 as was done for free testosterone and SHBG in supplemental figure 1? This would greatly help the readers interpret the clinical and biological significance of between group differences in hormone concentrations.

#6: Are the data shown in figure 6a and 6b repeated measures of the same samples? If so, it may be better to present the data as a time-series graph rather than as discrete bars on a bar graph.

Reviewer #3 (Remarks to the Author):

The article deals with an important question of sex specificity of the prevalence of the avian flu (H7N9) and associated mortality. It also touches upon characterization of the magnitude and heterogeneity of the endocrine and immune responses to this disease.

1. Avian flu caused by H7N9 virus is a devastating disease with mortality rate close to 40%. It disrupts a lot of bodily functions and perhaps the main reason one would focus attention on testosterone levels is to explain that the disease predominantly affects males. However, this is precisely the question the authors avoided to address.

2. As elsewhere in biology and medicine, one can discover myriad statistical associations among various observables. Many of them -- even quantitatively the most conspicuous ones -- are superficial because the observables may be highly dependent or driven by other factors beyond the scope of a particular study. The main goal of biomedical research is to uncover mechanisms and causal relationships, or at least formulate plausible falsifiable hypotheses about them that are supported by the collected data. The article under review didn't contribute much in this area. In particular, the article didn't shed light on the fundamental question regarding the following two non-mutually exclusive roles of testosterone in pathogenesis and the course of the disease caused by H7N9 virus in humans: (1) Does its low concentration make certain males more susceptible to the disease and more likely to die of it? or (2) Does the disease itself lead to the reduction in the testosterone levels?

3. In many data analyses conducted by the authors, a considerable fraction of observations (relative to sample size) was discarded as outliers based on a purely statistical criterion of large deviation from the interquartile interval. It is unclear whether this is justified; it is quite possible that these outliers contain information which is more important in certain respects than the bulk of the data. Crucially, some of the discarded outliers are males who both had H7N9 and high levels of testosterone, see e.g. Fig. 1a.

4. I believe that some of the data analyses miss an important step -- assessment of the extent of "natural" variation in the observed variables among controls. What are the measurement errors? What are the daily, other periodic, or perhaps lifestyle-related variations in the levels of testosterone and estradiol? The magnitude of such natural variation could serve as a scale for the assessment of the variation of measured observables in H7N9 (or H1N1 and H3N2) patients. Without this it is hard to ascribe strong reliability to the conclusions the authors draw from the collected data.

5. Concentrations of cytokines and chemokines form a system of highly dependent variables. Their univariate analysis may produce an incomplete, or even misleading, picture of the immune response to H7N9. A multivariate analysis would probably be more useful. Also, it is not obvious

that the cytokines and chemokines whose concentrations change most between survivors and non-survivors as well as between the two age groups are the most important ones.

6. I think the most important finding of the study is that H7N9 virus infects Leydig cells in mice and thereby reduces the level of testosterone. However, the discussion of this finding is too sketchy. What was previously known about this? Do other viruses e.g. SARS-CoV-1, SARS-CoV-2 and MERS have the same effect? What is the evidence that this virus-testosterone relationship can be translated from mice to men and if yes what are potential clinical implications?

7. I am concerned about statistical analysis conducted in the article, which serves as the sole basis for many of the article's conclusions. It is clear from the results that there are considerable differences between the distributions of the measured quantities among controls, see e.g. the boxplots in Fig. 1a for older males or in Fig. 1c for younger females. This violates distributional homogeneity, one of the critical assumptions behind any kind of statistical analysis.

8. What I am even more concerned about, however, is the use of p-values and statistical significance by the authors. They are applied ritualistically, i.e. without stating the null hypothesis, justifying the test statistic, verification of the underlying assumptions, and providing relevant context. P-values and statistical significance were introduced into empirical sciences by Ronald Fisher in the 1930s and became ubiquitously and uncritically employed since then in spite of numerous criticisms leveled against these statistical tools over the last hundred years. This has led to massive generation of false knowledge, especially in biomedical and social sciences. But only after it became clear over the last two decades that the majority of empirical scientific findings are irreproducible the statistical community realized the scale of misuse, abuse and misinterpretation of p-values, statistical significance and null hypothesis testing. For an official repudiation of such practices by the American Statistical Association, see

- Wasserstein, R.L.; Lazar, N.A. The ASA's statement on p-values: Context, process, and purpose. *The American Statistician* 2016, 70, 129–133.

- Wasserstein, R.L.; Schirm, A.L.; Lazar, N.A. Moving to a world beyond " $p < 0.05$ ". *The American Statistician* 2019, 73 (sup 1), 1-19.

The second article is an editorial to a volume that contains 43 articles some of which call for banishing p-values and statistical significance altogether. For a discussion of both general and technical aspects of the role of assumptions in statistical data analysis and another critical angle on p-values, see the following article:

- Hanin, L. Cavalier use of inferential statistics is a major source of false and irreproducible scientific findings. *Mathematics* (Basel, Switzerland) 2021, 9(6), article 603. doi:10.3390/math9060603

Point-by-point-Response

Reviewer#1:

I find the revised manuscript to be significantly improved. I have no additional concerns.

We thank the reviewer for his/her positive feedback.

Reviewer#2:

Overall, this is a well-organized study that strongly supports an association between low testosterone and increased influenza infection severity. This is a novel and significant finding in human patients that has previously only been shown with animal models of influenza infection. Moreover, the authors have now adequately addressed my concerns about cross-comparisons using different sample types and the added hospitalized seasonal influenza cohort greatly strengthens the relationship between low testosterone in males and severe outcomes with influenza.

We thank the reviewer for his/her positive feedback and constructive criticism.

Specific Points:

#1: However, the addition of Leydig cell infection data, while intriguing, is unconvincing and inadequate evidence is presented to support the conclusion that it is the direct infection of these cells that results in lower circulating testosterone. As a result, there are several questions and concerns that must be addressed if this data is to be included the current manuscript. Alternately, the Leydig cell data may be better saved for a future manuscript where the hypothesis can be fully developed and explored.

We agree with the reviewer and have now removed the Leydig cell data.

#2: The authors broadly conclude that it is the severe H7N9 infection that results in decreased male testosterone concentrations. However, the opposite relationship has also been demonstrated in murine models of influenza A virus infection (IAV) with low testosterone at the time of infection predisposing male mice to more severe outcomes with IAV infection. Given that both directional relationships may be true with H7N9 infection, and the lack of pre-infection testosterone concentration data, it seems as if only the conclusion that there is an association between low testosterone and increased severity H7N9 and seasonal influenza infection can be made in the current study. Likewise, were pre-infection testosterone concentrations measured for the mice in figure 6C? Given the high variability in testosterone concentrations, showing within mouse declines in testosterone following H7N9 infection would be more convincing than just showing a difference between the two treatment groups.

Thank you very much for this constructive suggestion. We have now performed new animal experiments to address the reviewer's valid concern. We now collected, **from the same animal**, blood samples 1 day **before infection** with H7N9 influenza virus, and then subsequently on day 1, 3 or 6 **post infection**, respectively (see new **Figure 6c-h**). For these analyses, we had to use 3 independent groups (pre-infection plasma and then from the same animal 1 day p.i.), (pre-infection plasma and then from the same animal 3 day p.i.) or (pre-infection plasma and then from the same animal 6 day p.i.). The reason behind this protocol is based on the animal welfare regulations and our obtained animal project license. For the measurement of hormone levels for each time point, we need 150µl blood. Within one week, we are only allowed to collect 2 blood samples per animal, which explains the study design. However, each data point shown in Figure

6c-h is obtained from the same animal, also explaining variations in the absolute values. We nevertheless chose to show absolute number instead of percentages to allow the reader an unbiased analysis of the data.

Obtained data from this new animal experiment, now further confirm that existing testosterone levels are further lowered upon H7N9 virus infection but also upon poly(I:C) treatment (see new **Figure 6c-h**). We also detected high virus load in the testes of H7N9 infected male mice as well as increased cytokine and chemokine levels in the testes of infected but not poly(I:C) treated mice (see new **Figure 6i-m**).

#3: Were cytopathic effects seen with the infection of the LC540 and primary Leydig cells? Was evidence of testicular inflammation or other pathological findings present in study participants or experimentally infected mice? If available, the inclusion of this data would greatly enhance the argument that it is the direct infection of these cells that results in the decreased systemic testosterone concentrations as opposed to the indirect effects inflammatory mediators have on the hypothalamic-pituitary-gonadal axis (doi.org/10.1210/jc.2015-3611 & doi.org/10.1089/107999099312948) demonstrated by others.

We followed the reviewer's suggestion and removed the Leydig cell culture data (see comment#1). However, we have now performed a new animal experiment, where we also measured virus titres and inflammation in the testes of the H7N9 infected mice. In the new **Figure 6b**, we show that H7N9 virus was detected after infection in testes and we measured testicular cytokine/chemokine levels on day 3 p.i., the time point of peak viral load. There, we detected an increase in testicular cytokine/chemokine response. However, gross pathology of the testes was not affected. These data suggest potential direct effects regarding the reduction of circulating testosterone levels in addition to likely indirect effects mediated by virus-induced inflammation acting on the HPG axis.

#4: Several of the box and whisker plots are missing one or both of their whiskers. This may just be an unfortunate artifact of the gaps added to the axes of these figures.

Generally, box and whisker plots need at least 5 calculated measures (min, 25%, 50%, 75%-quantile, max) to fully represent the classical graph. Hence, percentiles were missing in some figures due to the limited sample size in some subgroups in our study groups.

#5: Is it possible to include the reference ranges for total testosterone and estradiol in figures 1 and 2 as was done for free testosterone and SHBG in supplemental figure 1? This would greatly help the readers interpret the clinical and biological significance of between group differences in hormone concentrations.

Yes, we amended the graph accordingly.

#6: Are the data shown in figure 6a and 6b repeated measures of the same samples? If so, it may be better to present the data as a time-series graph rather than as discrete bars on a bar graph.

Leydig cell data were removed as suggested.

Reviewer#3:

The article deals with an important question of sex specificity of the prevalence of the avian flu (H7N9) and associated mortality. It also touches upon characterization of the magnitude and heterogeneity of the endocrine and immune responses to this disease.

We thank the reviewer for his/her feedback.

Specific Points:

#1: Avian flu caused by H7N9 virus is a devastating disease with mortality rate close to 40%. It disrupts a lot of bodily functions and perhaps the main reason one would focus attention on testosterone levels is to explain that the disease predominantly affects males. However, this is precisely the question the authors avoided to address.

With all due respect, we disagree. However, we hope that the revised version now with more causal data added, will be seen as a strong case by the reviewer. In addition we added more details from the epidemiological perspective of the data showing, that this paper is on retrospective and explorative screening of health care data, which restricts causal interpretation.

#2: As elsewhere in biology and medicine, one can discover myriad statistical associations among various observables. Many of them -- even quantitatively the most conspicuous ones -- are superficial because the observables may be highly dependent or driven by other factors beyond the scope of a particular study. The main goal of biomedical research is to uncover mechanisms and causal relationships, or at least formulate plausible falsifiable hypotheses about them that are supported by the collected data. The article under review didn't contribute much in this area. In particular, the article didn't shed light on the fundamental question regarding the following two non-mutually exclusive roles of testosterone in pathogenesis and the course of the disease caused by H7N9 virus in humans: (1) Does its low concentration make certain males more susceptible to the disease and more likely to die of it? or (2) Does the disease itself lead to the reduction in the testosterone levels?

We disagree with the reviewer that the current study is not shedding light on new perspectives and mechanisms regarding sex differences in infectious diseases.

Data obtained from the retrospective human data allowed the generation of new hypothesis, namely whether virus infection itself is mediating the reduction of testosterone levels. Therefore, as also suggested by reviewer#2, we performed animal studies to dissect potential causative role between infection and testosterone reduction (this reviewer's remark (2) in the last paragraph).

New Figure 6 now clearly shows that **pre-infection testosterone levels are lowered after H7N9 influenza virus infection**. We also detected a high virus load in the testes of the infected mice on days 3 and 6 p.i.. These data now strongly suggest direct (H7N9 virus detection in the testes of mice) as well as indirect (inflammation as shown by cytokine response in the sera of H7N9 infected humans and lung in mice) effects on the HPG axis resulting in low testosterone levels.

However, we generally agree with the reviewer's view that biomedical research is to uncover mechanisms and causal relationships. Overall, this is a multi-step process including animal

experiments, health care observations (secondary data use), clinical epidemiological observational and clinical experimental studies, respectively, which supports the grading scheme of evidence-based medicine. The **retrospective** data presented here, are from an observational health care surveillance setting. This data is therefore not from a clinical, i.e. experimental study contrasting well-defined groups following a randomization concept. Following this approach, all statistical tests do not support the proof of any causality. However, the explorative nature of the data may be used to develop hypotheses as we did here and with this are supported by the new data obtained from the animal model.

#3: In many data analyses conducted by the authors, a considerable fraction of observations (relative to sample size) was discarded as outliers based on a purely statistical criterion of large deviation from the interquartile interval. It is unclear whether this is justified; it is quite possible that these outliers contain information which is more important in certain respects than the bulk of the data. Crucially, some of the discarded outliers are males who both had H7N9 and high levels of testosterone, see e.g. Fig. 1a.

We respectfully disagree with this comment of the reviewer. At first, there was no formal rule of discarding outliers in the analysis. All box-plots were used using its standard Tukey's definitions in GraphPad Prism, which is out 1.5-fold of the inter-quartile-range is marked separately. Hence, all measured data were used. For all analyses comparing groups therefore non-parametric tests were performed, namely Wilcoxon for two and Kruskal-Wallis for three and more groups. Because no multi-mode data was identified these analyses should be identified as adequate. However, one argument of the reviewer may stated as relevant, which is, that some outliers contain information which is more important. In this revision, we added reference range for sex hormones, as suggested by reviewer#2, from related studies conducted from Chinese men and women (revised Figure 1 and 2) which may provide information of the distribution of values in general populations in China.

#4: I believe that some of the data analyses miss an important step – assessment of the extent of “natural” variation in the observed variables among controls. What are the measurement errors? What are the daily, other periodic, or perhaps lifestyle-related variations in the levels of testosterone and estradiol? The magnitude of such natural variation could serve as a scale for the assessment of the variation of measured observables in H7N9 (or H1N1 and H3N2) patients. Without this it is hard to ascribe strong reliability to the conclusions the authors draw from the collected data.

This is a generally valid question. Due to the nature of health care data, some of the important drivers are not available here and in addition, the data available does not support complex (generalized) regression analyses to take all factors into account. This is now discussed in detail in the discussion section as one restriction of the data. In addition, there are population-based reference ranges for the sex hormones. We have now included clinical references as suggested by reviewer#2 (see comment#5) and revised Figures 1 and 2 accordingly.

#5: Concentrations of cytokines and chemokines form a system of highly dependent variables. Their univariate analysis may produce an incomplete, or even misleading, picture of the immune response to H7N9. A multivariate analysis would probably be more useful. Also, it is not obvious that the cytokines and chemokines whose concentrations change most between

survivors and non-survivors as well as between the two age groups are the most important ones.

We fully agree with the reviewer that multivariate analysis is more useful to take the multicollinearity of the chemokines into account. However, this form of an analysis is not possible given the sample size. To further strengthen conclusions, we considered disease outcomes in the regression analysis between testosterone and cytokine/chemokine levels.

#6: I think the most important finding of the study is that H7N9 virus infects Leydig cells in mice and thereby reduces the level of testosterone. However, the discussion of this finding is too sketchy. What was previously known about this? Do other viruses e.g. SARS-CoV-1, SARS-CoV-2 and MERS have the same effect? What is the evidence that this virus-testosterone relationship can be translated from mice to men and if yes what are potential clinical implications?

We removed the Leydig cell data as suggested by reviewer#2. We extended the discussion regarding the impact of viral infections on the HPG axis.

#7: I am concerned about statistical analysis conducted in the article, which serves as the sole basis for many of the article's conclusions. It is clear from the results that there are considerable differences between the distributions of the measured quantities among controls, see e.g. the boxplots in Fig. 1a for older males or in Fig. 1c for younger females. This violates distributional homogeneity, one of the critical assumptions behind any kind of statistical analysis.

The distributional outcome of this study is indeed not homogeneous, which was the reason for non-parametric analyses, avoiding measures of location and the classical measurement error. We therefore re-phrase the material and method section dealing with the statistical analyses to make this more clear to the reader. Besides, we added reference ranges from Chinese general populations in revised Figure 1, which can help the reader to interpret the clinical and biological significance.

#8: What I am even more concerned about, however, is the use of p-values and statistical significance by the authors. They are applied ritualistically, i.e. without stating the null hypothesis, justifying the test statistic, verification of the underlying assumptions, and providing relevant context. P-values and statistical significance were introduced into empirical sciences by Ronald Fisher in the 1930s and became ubiquitously and uncritically employed since then in spite of numerous criticisms leveled against these statistical tools over the last hundred years. This has led to massive generation of false knowledge, especially in biomedical and social sciences. But only after it became clear over the last two decades that the majority of empirical scientific findings are irreproducible the statistical community realized the scale of misuse, abuse and misinterpretation of p-values, statistical significance and null hypothesis testing. For an official repudiation of such practices by the American Statistical Association, see

- *Wasserstein, R.L.; Lazar, N.A. The ASA's statement on p-values: Context, process, and purpose. The American Statistician 2016, 70, 129–133.*
- *Wasserstein, R.L.; Schirm, A.L.; Lazar, N.A. Moving to a world beyond “ $p < 0.05$ ”. The American Statistician 2019, 73 (sup 1), 1-19.*

The second article is an editorial to a volume that contains 43 articles some of which call for banishing p-values and statistical significance altogether. For a discussion of both general and technical aspects of the role of assumptions in statistical data analysis and another critical angle on p-values, see the following article:

Hanin, L. Cavalier use of inferential statistics is a major source of false and irreproducible scientific findings. Mathematics (Basel, Switzerland) 2021, 9(6), article 603. doi:10.3390/math9060603

As mentioned in our answer to comment #2, our data is from an observational study of health care data, which prohibits any causal interpretation of the statistical tests. In addition to Fisher's start at the beginning of the 20th century and his 5% definition (with which some members of the Royal Statistical Society were not happy with), a lot of discussions about p-values were made since then with John Ioannidis' scientific waste debate as one highlight of the p-value's common misinterpretation in medicine. Some authors avoid to use statistical tests in observational data completely due to the nature of statistical tests, i.e. one hypothesis – one test – one p-value.

We therefore have to apologize our strict wording, which appear to be ritualistically. We strictly re-phrase our manuscript in this sense and give additional discussions on this topic to avoid this wrong interpretation.

Reviewer comments, further round review

Reviewer #2 (Remarks to the Author):

The authors have addressed all my previous concerns apart from the discussion section conclusion on Lines 478-480 regarding locally driven suppression of testicular testosterone production. It is still not clear from the presented data that the suppression of testosterone is the direct result of testicular infection. As both nasal inoculation with poly(I:C) and H7N9 resulted in equivalent levels of testosterone suppression, the presence of virus and increased testicular cytokine concentration does not exclude the possibility that testosterone concentrations are being suppressed through systemic actions on the HPG axis. Especially given the lack of observed testicular histopathology. This is a minor objection however that can be easily addressed through word smithing and does not necessitate another round of review.

Reviewer #3 (Remarks to the Author):

As a result of substantial revision, the article has seen a major improvement in terms of depth, content and clarity. In particular, new experiments conducted by the team of authors have brought them a step closer to understanding a complex causal relationship between H7N9 infection and low testosterone levels in males.

A few minor technical and stylistic comments on outstanding issues are given below.

1. Figures 5 and S3. What is the null hypothesis and test statistic for which p-values were computed?
2. Line 138. "status" should probably be changed to "state".
3. Line 191. What are the units for 0.1?
4. Line 195. The correct name of the "Madine Darby Canine Kidkey" cell line is "Madin-Darby Canine Kidney".
5. Lines 256-257. Study participants included both males and females, so universal interpretation of age brackets in terms of menopausal status does not make sense.
6. Line 273. "variance" should probably be changed to "variation".
7. Line 562. I believe "testosterone" should be changed to "estradiol".
8. Many phrases in the manuscript, too numerous to be listed individually, are awkward and hard to interpret. Examples are: "The high degree of cases towards men" (lines 62-63), "we have divided the mouse of the experiment" (lines 156-157), "values without normal distribution were transformed into natural logarithm" (line 245), "analysis of the higher amount in male H7N9 infections", "interaction between death and survivors" (line 613), etc. Also, the manuscript contains many grammatical/semantic errors. I believe the style of the article will benefit greatly from being edited by a native English speaker.
9. In some references, the first letters of the words in the title are capitalized (see refs 11, 14, 27, 32, 34, 35) while in others they are not.

Point-by-point response to reviewer comments on manuscript NCOMMS-20-22118B-Z from September 28th 2022

Reviewer #2

1. The authors have addressed all my previous concerns apart from the discussion section conclusion on Lines 478-480 regarding locally driven suppression of testicular testosterone production. It is still not clear from the presented data that the suppression of testosterone is the direct result of testicular infection. As both nasal inoculation with poly(I:C) and H7N9 resulted in equivalent levels of testosterone suppression, the presence of virus and increased testicular cytokine concentration does not exclude the possibility that testosterone concentrations are being suppressed through systemic actions on the HPG axis. Especially given the lack of observed testicular histopathology. This is a minor objection however that can be easily addressed through word smithing and does not necessitate another round of review.

We thank the reviewer for his/her positive feedback. We have amended the respective section in the discussion accordingly.

Reviewer #3

As a result of substantial revision, the article has seen a major improvement in terms of depth, content and clarity. In particular, new experiments conducted by the team of authors have brought them a step closer to understanding a complex causal relationship between H7N9 infection and low testosterone levels in males.

We thank the reviewer for his/her positive feedback.

A few minor technical and stylistic comments on outstanding issues are given below.

1. Figures 5 and S3. What is the null hypothesis and test statistic for which p-values were computed?

The null hypothesis is “correlation=0”. In Figure 5, we computed p-values for the test of correlation between overall testosterone and cytokines in H7N9 cases (including both deaths and survivals, displayed in each figure). We also computed p-values for the interaction of the disease outcomes and we didn’t find any interaction between survival and death (data not shown). In Figure S3, we computed the p-values of the correlation of testosterone and cytokines in seasonal influenza outpatients or hospitalized respectively.

2. Line 138. “status” should probably be changed to “state”.

Yes, we corrected this accordingly.

3. Line 191. What are the units for 0.1?

The unit is pg/ml in human samples for cytokine/chemokine measurement and ng/ml for murine lung cytokine/chemokine measurement, which we have now amended.

4. Line 195. The correct name of the “Madine Darby Canine Kidkey” cell line is “Madin-Darby Canine Kidney”.

Yes, we corrected this accordingly.

5. Lines 256-257. Study participants included both males and females, so universal interpretation of age brackets in terms of menopausal status does not make sense.

Yes, we amended this as suggested.

6. Line 273. “variance” should probably be changed to “variation”.

Yes, we corrected this accordingly.

7. Line 562. I believe “testosterone” should be changed to “estradiol”.

Yes, we corrected this accordingly.

8. Many phrases in the manuscript, too numerous to be listed individually, are awkward and hard to interpret. Examples are: “The high degree of cases towards men” (lines 62-63), “we have divided the mouse of the experiment” (lines 156-157), “values without normal distribution were transformed into natural logarithm” (line 245), “analysis of the higher amount in male H7N9 infections”, “interaction between death and survivors” (line 613), etc. Also, the manuscript contains many grammatical/semantic errors. I believe the style of the article will benefit greatly from being edited by a native English speaker.

Yes, we agree. The manuscript was now edited for language by the Nature editing service.

9. In some references, the first letters of the words in the title are capitalized (see refs 11, 14, 27, 32, 34, 35) while in others they are not.

Yes, we corrected this accordingly.